# RELIABLE EVALUATION OF MRI MOTION CORREC-TION: DATASET AND INSIGHTS

**Kun Wang**[*], **Tobit Klug**[*], **Stefan Ruschke**[†], **Jan S. Kirschke**[†], **and Reinhard Heckel**[*,⋆]
[*]School of Computation, Information and Technology, Technical University of Munich
[†]School of Medicine and Health, Technical University of Munich
[⋆]Munich Center for Machine Learning

## ABSTRACT

Correcting motion artifacts in scientific and medical imaging is important, as they significantly impact image quality. However, evaluating deep learning-based and classical motion correction methods remains fundamentally difficult due to the lack of accessible ground-truth target data. To address this challenge, we study three evaluation approaches: real-world evaluation based on reference scans, simulated motion, and reference-free evaluation, each with its merits and shortcomings. To enable evaluation with real-world motion artifacts, we release PMoC3D, a dataset consisting of unprocessed **P**aired **Mo**tion-**C**orrupted **3D** brain MRI data. To advance evaluation quality, we introduce MoMRISim, a feature-space metric trained for evaluating motion reconstructions. We assess each evaluation approach and find real-world evaluation together with MoMRISim, while not perfect, to be most reliable. Evaluation based on simulated motion systematically exaggerates algorithm performance, and reference-free evaluation overrates oversmoothed deep learning outputs. Overall, these contributions advance the emerging science of benchmarking for machine learning based scientific and medical imaging, by providing datasets, metrics, and systematic evaluations for motion correction.

## 1 INTRODUCTION

Magnetic resonance imaging (MRI) provides rich anatomical detail but requires long acquisition times. Even healthy adults struggle to remain still for several minutes, and small head movements introduce inconsistencies in k-space that lead to blurring, ringing, and spatial misalignment in the reconstructed images. Motion artifacts can prevent effective diagnosis and may force repeating the scan (Andre et al., 2015; Slipsager et al., 2020), increasing costs and reducing the reliability of clinical workflows. Robust motion correction is therefore essential for ensuring dependable MRI acquisition.

Many recent approaches tackle motion correction retrospectively, estimating and correcting motion directly from the acquired k-space or reconstructed images without relying on external motion-tracking hardware. Several works (Haskell et al., 2019; Singh et al., 2022; 2023; Levac et al., 2023) proposed methods for reconstructing 2D motion-corrupted slices. However, subject motion occurs in full 3D, making the problem more challenging and clinically relevant. Prior work has shown promising 3D motion-estimation performance (Johnson & Drangova, 2019; Duffy et al., 2021; Klug et al., 2024). (Cordero-Grande et al., 2016) introduced a classical alternating-optimization framework for jointly estimating motion and reconstructing corrupted volumes. (Al-Masni et al., 2022) proposed a stacked U-Net that performs end-to-end 3D reconstruction without explicit motion modeling. More recently, (Wu et al., 2025) presented an implicit neural representation method that jointly estimates motion and reconstructs images in both 2D and 3D MRI.

However, research on 3D motion correction is challenging as the field lacks a standardized evaluation approach for realistically evaluating different approaches. The core issue is that ground-truth target data is fundamentally difficult or impossible to obtain:

- Real-world motion-corrupted data captures true motion but lacks ground-truth for quantitative assessment. To enable real-world evaluation, one can collect two scans, a motion-

corrupted and a motion-free one, and use the motion-free as a target or ground-truth. However, the two scans need to be aligned, require careful pre-processing, and the motion-free scan might also be slightly motion corrupted.

- Most commonly, evaluation is conducted based on simulated rigid motion artifacts in which case computing reference-based metrics is straightforward. However, evaluation might be unrealistic due to the motion simulation. Potential artifacts from non-rigid motion are not accounted for when simulating motion (Spieker et al., 2024). In addition, data has to be fully sampled to simulate motion, which is rarely the case for 3D MRI.

- Finally, evaluation can be conducted with reference-free image quality metrics, which avoid the need for pre-processing or the lack of a realistic reference. However, classical gradient-based metrics correlate poorly with perceived image quality (Marchetto et al., 2024).

In this work, we advance evaluation of motion reconstruction algorithms by systematically assessing real-world, simulated, and reference-free evaluation, and by providing PMoC3D, a real-world dataset for evaluation of 3D-motion correction methods. Our dataset, PMoC3D, consists of raw **P**aired **Mo**tion-**C**orrupted **3D** brain MRI data for enabling real-world evaluation by comparing reconstructions motion-corrupted data to a reference scan. For PMoC3D, we collected three motion-corrupted scans of different motion severity each from eight subjects along with one motion-free scan to use as a reference.

First, we study real-world evaluation with a reference scan based on the PMoC3D dataset by assessing how well reference-based image quality metrics correlate with human assessment. We consider standard metrics in the pixel-space such as SSIM (Wang et al., 2004) and PSNR (Horé & Ziou, 2010) and feature-space metrics such as DreamSim (Fu et al., 2023), and we additionally propose a feature-space metric MoMRISim that is trained to align with varying levels of motion severity. We find that reference-based evaluation using feature-space metrics like MoMRISim correlates well with human judgments and provides a reliable measure of reconstruction quality. However, under mild motion, the motion-free reference reconstructions often retain residual artifacts, and in some cases, mildly motion-corrupted scans reconstructed with motion-correction methods appear visually cleaner than the reference. This challenges the reliability of reference-based evaluation in mild motion settings, where simulated data with known ground truth can offer a more meaningful alternative for evaluation.

Second, we assess evaluation based on simulated motion corruption, and observe that some methods achieve almost error free reconstructions under the most severe simulated motion, whereas the same methods exhibit noticeable residual artifacts under severe real-world motion. This is consistent with findings for other imaging problems, that found simulated data to potentially lead to misleading conclusions (Shimron et al., 2022).

Third, regarding reference-free evaluation, we propose and consider a vision-language model (VLM) score. While exhibiting a significantly better alignment with perceived image quality than classical gradient-based reference-free metrics, we find the VLM score to be biased towards reconstructions, which are overly smooth but potentially miss anatomical details.

All three considered evaluation methods have shortcomings, but evaluation on real-world paired datasets such as PMoC3D, when combined with an appropriate feature-based metric such as MoMRISim, provides a relatively reliable and meaningful assessment of reconstruction performance under moderate to severe motion. Overall, these contributions advance the emerging science of benchmarking for machine learning based scientific and medical imaging, by providing datasets, metrics, and systematic evaluations for motion correction.

## 2 THE PMOC3D DATASET FOR REAL-WORLD EVALUATION

We constructed the PMoC3D dataset in order to evaluate accelerated 3D motion correction methods. PMoC3D is a 3D dataset containing the raw measurement data of scans with real-world motion as well as corresponding motion-free scans as a target or estimate of the ground-truth.

Previous works relied on evaluation datasets that provide only the processed magnitude images (Johnson & Drangova, 2019; Duffy et al., 2021; Ganz & Eichhorn, 2022; Li et al., 2024).

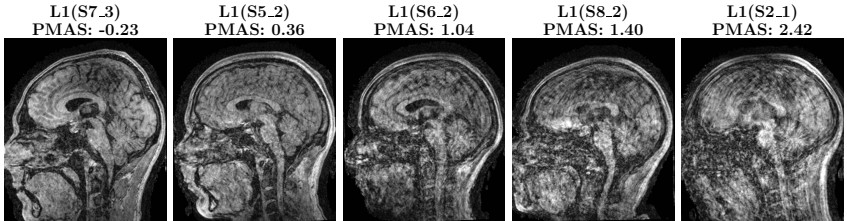

Figure 1: Sagittal views and corresponding perceived motion artifact scores (PMAS) of selected L1-reconstructions from our dataset with varying degrees of motion artifacts ranging from mild (**S7_3**, **S5_2**) to moderate (**S6_2**) to severe (**S8_2**, **S2_1**). These examples highlight the challenges in reconstructing motion-corrupted scans.

However, approaches that explicitly estimate motion in order to correct for it, require access to the unprocessed raw data (k-space data) (Cordero-Grande et al., 2016; Klug et al., 2024).

## 2.1 ACQUIRING PAIRED MOTION-FREE AND MOTION-CORRUPTED DATA

For the PMoC3D dataset we scanned 8 healthy male and female subjects. The study was approved by the institutional review board and written informed consent was obtained from all participants prior to data collection (see Ethical Statement at the end of the paper for details). From each subject we acquired four scans, one motion-free and three motion-corrupted labeled as S{subject}_{scan} with subjects $1, \ldots, 8$ and scans $0, \ldots, 3$, where 0 corresponds to the motion-free case. When we observed artifacts in the scanner's reconstruction while recording a motion-free scan, we assumed that the subject has moved and restarted the scan session. We provide access to the corrupted scans (S3_4, S5_4, S8_4) with involuntary motion resulting in a total of 27 motion-corrupted scans.

**Data acquisition.** The data was acquired on an Ingenia Elition 3.0T X scanner (Philips Healthcare, Best, The Netherlands) with the standard 16-channel dStream HeadSpine coil array (of which 13 channels were automatically selected based on SNR). We performed Cartesian 3D T1-weighted Fast-Gradient-echo (TFE) imaging with a 1mm isotropic resolution and a field of view of $221 \times 170 \times 256$ mm. With $k_z$ and $k_x$ oversampling of 1.4 and 2.0, the acquisition matrix-size is $k_y \times k_z \times k_x = 222 \times 236 \times 512$.

Data was acquired with undersampling along the phase encoding dimensions $k_y \times k_z$ (axial plane) with an undersampling factor of $\mu = 4.94$, a densely sampled auto-calibration region of size $37 \times 37$ and partial-Fourier sampling with factor 0.85 in the $k_z$ direction(see Figure 5 in Appendix B for the resulting undersampling mask).

We provide access to the full-size k-space data. To reduce the computational cost for evaluating experiments, we also cropped the data, where we crop along the fully-sampled read-out dimension $k_x$ to the size of the field of view (256) by subsampling every second voxel. The sequence parameters are in Appendix B.

**Sampling trajectory.** The Cartesian k-space is acquired within $N_s = 52$ shots resulting in $k_y * k_z/(\mu N_s) = 222 * 236/(4.94 * 52) = 204$ acquired read-out lines per shot. The acquisition of one shot lasts 1.35s followed by a pause of 1.74s resulting in a total scan duration of 2:40min. The read-out lines are acquired following a quasi-random sampling trajectory except the $3 \times 3$ center of the k-space which is acquired at the start of the first shot. Hence, the sampling order varies between all scans in the dataset.

We chose a random order because it ensures that both low-/high-frequency components are sampled in every shot which is beneficial for motion estimation (Cordero-Grande et al., 2016; Usman et al., 2020; Klug et al., 2024).

**Sensitivity maps.** For each subject, the dataset contains the coil sensitivity maps calculated using Gyrotools MRecon (LLC) through a calibration scan performed at the beginning of each subject's scan session for which the subject was instructed to hold still.

Table 1: Overview of real-world paired, simulated motion, and reference-free evaluation.

| Evaluation Method | Real-world Paired | Simulated Motion | Reference-Free |
|---|---|---|---|
| Data Source | PMoC3D | Calgary Campinas Brain MRI dataset (Souza et al., 2018) | PMoC3D |
| Reference Type | Paired reference-free scan | Perfect ground truth | None |
| Eval Metrics | PSNR, SSIM, DISTS MoMRISim, ... | PSNR, SSIM | TG, AES, VLM scores |
| Human Alignment | High | – | Low |
| Limitations | Imperfect reference-free scan | Simulation fails to reproduce real motion complexity | • Weak correlation with human evaluation
• Favors oversmoothed images |
| Conclusion | Not perfect, but most reliable | Overestimate algorithm performance | Unreliable |

**Motion.** Motion corrupted scans are obtained by instructing the subject to perform a motion at one or more time instances during the scan. The instructions are as follows:

- Slightly turn head left/right and stay/return to origin.
- Nodding: Once up, once down and return to origin.
- Head shaking: Once left, once right and return to origin.
- Move chin towards chest and stay/return to origin.

We instruct to perform a motion slowly if it should be performed slowly as opposed to abruptly. To generate a diverse dataset containing mild to severe motion artifacts we vary the instructions itself, when we give them, and how many we give (up to three). The instructions and time stamps are provided with the data.

## 2.2 CATEGORIZING MOTION ARTIFACTS IN THE DATA

Stronger motion and more motion events make reconstruction more difficult. To facilitate evaluation with the PMoC3D dataset, we provide a quantitative measure of human-perceived severity of the motion artifacts of each of the 24 scans corrupted with voluntary motion.

The score is computed as follows. We first obtain the L1-reconstruction with wavelet regularization (Lustig et al., 2008) without any motion correction for each scan (see Appendix D.1 for details). The L1-reconstruction with wavelet regularization does not account for motion modeling or correction, allowing us to directly evaluate the severity of motion artifacts in each scan.

Two PhD students with expertise in machine learning and MRI reconstruction performed pairwise comparisons between the reconstructions, classifying either one to have more or both to have similarly severe artifacts. Since PMAS here is used only to categorize the severity of motion artifacts in the raw scans, the use of trained non-clinical evaluators is sufficient for this purpose. If both evaluators agree that reconstruction A has more severe artifacts than B then we assign rate $p(A > B) = 1$. If one evaluator judges A to be better and the other finds a similar level, then we assign $p(A > B) = 0.75$. If both evaluators find a similar severity level, or one finds one better and the other the other, we set $p(A > B) = 0.5$.

Based on these pairwise results we fitted a Bradley–Terry model (Bradley & Terry, 1952) to obtain a perceived motion artifact score for each scan:

$$\text{PMAS} = \arg\max_{\beta} \sum_{i \neq j} p(i > j) \log \left( \frac{\exp(\beta_i)}{\exp(\beta_i) + \exp(\beta_j)} \right), \tag{1}$$

where each $\beta_i$ quantitatively represents the severity of motion artifacts for the corresponding volume; higher values indicate more severe artifacts, and these latent parameters serve as our measure of perceived motion artifact severity(see Appendix C for more details).

Figure 1 shows sagittal views of L1-reconstructions from our dataset from left to right in ascending order according to their perceived motion artifact score. As we can see, the severity of motion artifacts increases with increasing perceived motion artifact score. Reconstructions S7_3 and S5_2 show mild artifacts, where most brain anatomical details are preserved despite the presence of minor ringing artifacts. In reconstruction S6_2, the artifacts are more pronounced and obscure finer details, while in S8_2 and S2_1 the artifacts are severe enough that the brain structures become barely discernible. These examples illustrate the range of challenges encountered when reconstructing motion-corrupted scans in our dataset.

## 3   EVALUATION APPROACHES

Evaluating motion correction in 3D MRI is challenging because we do not have the perfect reference. We have considered 3 evaluation methods, each with its own strengths and failure modes. Table 1 provides an overview comparison of the three evaluation methods considered in this work.

In the remainder of this section, we describe these three evaluation approaches in detail: real-world evaluation with a reference scan, evaluation based on simulated motion, and reference-free evaluation. We also discuss standard evaluation metrics and propose two novel metrics for the respective approaches.

### 3.1   REAL-WORLD EVALUATION WITH A REFERENCE SCAN

We perform real-world evaluation with our PMoC3D dataset, which consists of paired acquisitions for each subject. Each subject Si (i=1,...,8) undergoes one motion-free scan Si_0 and three motion-corrupted scans Si_j (j=1,2,3). The motion-corrupted scans are categorized into two difficulty levels: the 8 scans with the lowest perceived motion artifact scores (PMAS) are labeled as mild motion-corrupted scans, while the remaining scans are classified as moderate and severe. Each baseline method is applied to each motion-corrupted scan, and the L1 reconstruction of the motion-free scan is the reference for quantitative scoring.

We preprocess the data to mitigate the challenges of comparing two different acquisitions, as suggested by the paper (Marchetto et al., 2024). First, using advanced normalization tools(ANTs) (Tustison et al., 2021) rigidly aligns the motion-corrupted volume to the reference. Subsequently, a brain mask is generated via BET (Smith, 2002) on the motion-free scan and applied to both datasets, to focus the evaluation on the anatomical region of interest. Both volumes are multiplied by the brain mask and normalized to a max value of the 99.9th percentile before computing scores. After preprocessing, reference-based quality metrics are computed between the corrected motion-corrupted scans and the L1-based motion-free reference reconstructions (Si_0).

### 3.2   EVALUATION BASED ON SIMULATED MOTION

We evaluate performance by generating synthetic motion-corrupted, undersampled k-space data from the fully sampled Calgary Campinas Brain MRI dataset (Souza et al., 2018), and comparing reconstructions against the original, uncorrupted reference volumes. This has the advantage that we have accurate ground-truth or target information, and the disadvantage that the motion is synthetic.

A 3-D Cartesian mask with an acceleration factor of 4.9 is applied in the two phase-encoding directions, replicating the mask geometry used for PMoC3D. Each acquisition is divided into 52 shots following a random trajectory, again mirroring the paired real-world protocol. Inter-shot head motion is generated with an event-based framework designed to resemble PMoC3D artifacts. Motion events follow the instructions in Section 2 (head turning, nodding, etc.) which involve rotations about the $k_y$-$k_z$ and $k_x$-$k_z$ axes. To more realistically capture head motion, we introduce random perturbations to the remaining motion parameters, i.e., the three translational components and the third rotational axis. These perturbations account for natural subject-specific variability and for the fact that real-world head rotations often occur around off-center axes rather than the image origin, resulting in complex motion patterns. Motion severity is controlled by the number of events and their amplitude:

- **Mild**: One event with primary motion sampled uniformly from $\pm 5°$ and perturbations up to $\pm 1°$/mm.

- **Severe**: Three events with primary motion sampled uniformly from $\pm 15°$ and perturbations up to $\pm 5°$/mm.

To match the behavior of our real acquisitions, the simulation parameters are chosen to mirror those in PMoC3D. Mild motion contains a single event, and severe motion contains three events, consistent with the PMoC3D acquisition protocol. The motion amplitudes were selected based on visual inspection of the real severe L1 reconstructions to ensure that the simulated artifacts are at least as challenging as the most corrupted in-vivo scans.

For each baseline, we evaluate performance across ten test volumes. For every volume, we simulate the two severity levels, and for each level, we draw two independent motion seeds. Reconstructed volumes are normalized to their 99.9th percentile intensity, after which reference-based metrics are computed against the fully sampled ground-truth images. While simulated motion offers controllable and repeatable conditions, it may oversimplify the motion dynamics in vivo.

### 3.3 REFERENCE-FREE EVALUATION

In the reference-free setting, each baseline method reconstructs the motion-corrupted scans Si_j, for i in 1,...,8, and j from 1 to 3 on PMoC3D. The reconstructed volumes are first masked to exclude non-brain regions, consistent with the paired evaluation protocol. Subsequently, the volumes are then normalized by the 99.9th percentile normalization, matching the normalization used in the paired evaluation. Because the available motion-free scans cannot serve as true ground truth, the normalised reconstructions are evaluated directly with a reference-free metric discussed in the next section.

### 3.4 EVALUATION METRICS

We first review existing reference-based and reference-free metrics, then introduce our two proposed metrics: MoMRISim (reference-based) and the VLM score (reference-free).

#### 3.4.1 EXISTING METRICS

**Reference-based metrics.** We consider standard pixel-wise metrics including Structural Similarity Index (SSIM) (Wang et al., 2004), Peak Signal-to-Noise Ratio (PSNR) (Horé & Ziou, 2010), and Artifact Power (AP) (Ji et al., 2007). We also consider perceptual metrics, which evaluate image quality based on high-level visual representations rather than direct pixel-wise comparisons. These metrics leverage deep learning-based feature extraction to assess structural and perceptual similarity, and have demonstrated strong correlation with radiologists' assessments (Adamson et al., 2025). Specifically, we consider Deep Image Structure and Texture Similarity (DISTS) (Ding et al., 2022) and DreamSim (Fu et al., 2023).

**Reference-free metrics.** We also investigate reference-free metrics for assessing image quality without requiring access to a motion-free reference scan. We employ two gradient-based methods, Average Edge Strength (AES) (Pannetier et al., 2016) and Tenengrad (TG) (Kecskemeti et al., 2018), which quantify image sharpness and structural clarity.

#### 3.4.2 MoMRISim FOR REFERENCE-BASED EVALUATION

Existing image-quality metrics struggle to capture motion artifacts in 3D MRI. Pixel-based metrics such as PSNR and SSIM exhibit only moderate correlation with human evaluation. Perceptual metrics are trained to model semantic similarity in natural images, rather than the motion artifacts in MRI. Moreover, adapting perceptual metrics like DreamSim (Fu et al., 2023) typically requires large-scale human annotations, which are impractical to collect for subtle 3D motion artifacts. To overcome these limitations, we propose **MoMRISim**, a perceptual similarity metric tailored specifically to motion artifacts in 3D MRI. Unlike DreamSim, which is trained on human-labeled natural-image triplets, MoMRISim utilizes motion severity as a self-supervised signal and is therefore optimized directly for motion artifact discrimination.

Our goal is to learn an encoder $f(\cdot)$ whose feature space reflects motion severity through its distance to the motion-free reference. In this space, mildly corrupted reconstructions should lie closer to their

reference than severely corrupted ones. To obtain such an encoder, we train it using triplets consisting of a motion-free reference and two motion-corrupted reconstructions with known simulated severities. Because the severity ordering in each triplet is exact by construction, the model receives perfect relative supervision without requiring any human annotations. The encoder is optimized so that mildly corrupted images are embedded closer to the reference than their more severely corrupted counterparts. Triplets are constructed by applying synthetic rigid motion of varying severity, defined following the protocol in paper (Klug et al., 2024), to the fully sampled Calgary Campinas Brain MRI dataset (Souza et al., 2018). To enhance robustness across reconstruction styles, we randomly apply either L1- or U-Net-based reconstruction without motion correction, encouraging the encoder to learn motion–artifact features that are invariant to the two reconstruction pipelines used in this study.

At evaluation time, the triplet structure is no longer needed: MoMRISim reduces to a standard reference-reconstruction pair. MoMRISim assigns a motion severity score to reconstruction $X$ relative to its reference $R$ by computing the cosine distance between their embeddings:

$$\text{MoMRISim}(R, X) = 1 - \text{CosineSimilarity}\big(f(R), f(X)\big),$$

where $f(\cdot)$ is the learned feature extractor. Higher values indicate greater deviation from the reference and, therefore, more severe motion artifacts. Full training details, including data preparation and hyperparameters, are provided in Appendix E, with Figure 6 illustrating a triplet-based input example used during training.

### 3.4.3 VLM SCORE FOR REFERENCE-FREE EVALUATION

Classical gradient-based image quality metrics have been shown to correlate poorly with human judgments in prior work (Marchetto et al., 2024). Our own experimental results in Appendix G.1 corroborate this finding, highlighting the limitations of these metrics in evaluating motion artifacts in 3D MRI. To address this, we propose a reference-free **VLM score** based on prompting a vision-language model. We evaluate our approach using GPT-4o (OpenAI, 2024b;a), Qwen2.5-VL-Max (Bai et al., 2025), Med3DVLM (Xin et al., 2025), and M3D-LaMed (Bai et al., 2024). The models are asked to assign motion artifact severity score ranging from 0 (no motion) to 3 (severe motion). To enhance robustness, we prompt the model independently five times at a temperature of 0.5 and compute the final score as the average across all runs. See Appendix F for details.

## 4 ASSESSING EVALUATION APPROACHES

In this section, we assess the three evaluation approaches.

### 4.1 IMPLEMENTATION DETAILS AND BASELINES

We evaluate three motion-reconstruction methods that span the major paradigms used in 3D MRI motion correction: a classical optimization-based method that relies purely on physics, a hybrid deep-learning and physics approach, and a fully end-to-end deep learning model that corrects motion without explicit motion modeling. Covering these three categories allows us to assess metric performance across fundamentally different reconstruction strategies. Following three motion reconstruction methods are utilized to generate reconstructions for evaluation:

- The classical *alternating optimization* (AltOpt) (Cordero-Grande et al., 2016) alternately optimizes the L1–wavelet reconstruction and the motion parameters, updating one while holding the other fixed.

- The deep learning-based *MotionTTT* (Klug et al., 2024) relies on a 2D U-net (Ronneberger et al., 2015) pre-trained to perform motion-free MRI reconstruction. MotionTTT estimates the 3D motion parameters by optimizing a data-consistency (DC) loss between the network output and the given motion-corrupted measurements over the motion parameters.

- *E2E Stacked U-nets* (Al-Masni et al., 2022) is based on a stack of refinement U-nets to predict the motion-corrected reconstruction slice-wise. The training details are described in Appendix D.4.

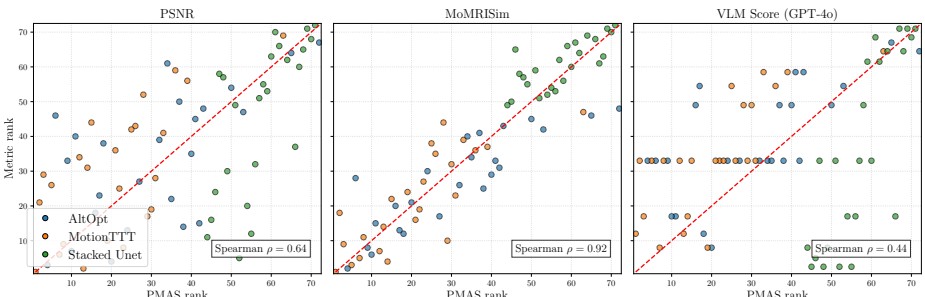

Figure 2: Rank comparison between PSNR, MoMRISim, and VLM score (GPT-4o) and the perceived motion artifact score (PMAS). Both axes show rank values rather than raw scores. A smaller rank indicates a better reconstruction. Points lying closer to the diagonal reflect stronger agreement between a metric's ordering and the PMAS ordering. MoMRISim shows a strong correlation, PSNR offers a moderate level, and VLM score(GPT-4o) reflects a low degree of alignment.

MotionTTT and stacked U-nets are both trained on the Calgary Campinas Brain MRI Dataset (Souza et al., 2018) consisting of T1-weighted 3D motion-free brain scans recorded under a similar setting as our PMoC3D dataset. For MotionTTT and AltOpt, we estimate six motion parameters (three rotations, three translations) per acquired shot, and perform L1-minimization based on the estimated motion parameters for the final reconstruction. Shots with motion parameters that have a data consistency (DC) loss above a certain threshold are excluded from the measurements. See Appendix D for all training details and hyperparameters.

## 4.2 Assessing paired real-world evaluation

Paired real-world evaluation is challenging because no truly motion-free ground truth exists. In PMoC3D, the reference volume is acquired in a separate scan and reconstructed via L1-minimization, and thus is not perfect ground-truth data. Moreover, it might be slightly misaligned to the motion-corrupted scan even if the motion-free scan had no motion corruption.

**Assessing evaluation reliability with a human judge.** Because the motion-free volume is not a perfect ground truth, we evaluate reliability by studying whether the ranking produced by different metrics measuring similarity of reconstructions and references based on the PMoC3D dataset agree with human perception.

We performed pairwise comparisons of 250 randomly selected baseline reconstruction pairs, with a licensed medical doctor judging whether one image exhibited more severe artifacts or both had similar severity. The resulting preference matrix was fitted with a Bradley-Terry model (details in Appendix D.1), which yields perceived motion artifact scores (PMAS). To verify the reliability of PMAS ratings, we also collected a full set of 2,556 pairwise comparisons from a second rater for inter-rater analysis (which was not used in any subsequent experiments). The resulting PMAS ranks show strong consistency; see Appendix G.5 for details.

We assessed the association between PMAS and each evaluation metric using Spearman's rank correlation coefficient. Correlations for PSNR and MoMRISim are displayed in Figure 2; results for all metrics appear in the Appendix G.1. Figure 2 shows that PSNR and MoMRISim yield rankings consistent with the human judgment (i.e., with the perceived motion artifact scores (PMAS)). MoMRISim, in particular, attains the highest correlation, which is 0.92. This strong association demonstrates that our evaluation is a faithful proxy for expert assessment, confirming the reliability of paired real-world evaluation.

**Challenges in the paired real-world evaluation in particular for mild motion.** Figure 3 shows the motion-free reference image and the MotionTTT reconstruction of scan S3_3, which was acquired under mild subject motion, along with the difference of the two images. For this example, the MotionTTT reconstruction of the mild motion scan is slightly better at parts than the reference image, which can be seen in the zoomed region: The MotionTTT reconstruction is free of ringing,

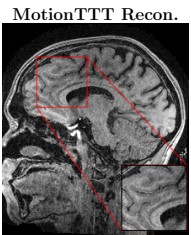 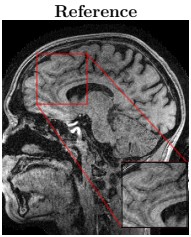 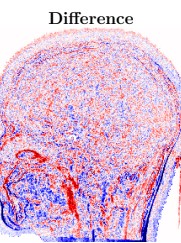

Figure 3: Comparison of MotionTTT reconstruction, L1-based motion-free reference, and their difference image. The left panel shows the MotionTTT reconstruction of subject S3_3, which is a mild motion scan. The center panel displays the L1 reconstruction of the corresponding motion-free reference. The right panel presents the difference image between the two reconstructions.

whereas the L1 reconstructed motion-free reference contains faint ring artifacts. Under mild motion, the corrected image can therefore look better than the reference, which is problematic.

To summarize, although imperfect reference volumes may render paired real-world evaluation empirically unreliable in the mild-motion regime, real-world evaluation is dependable under moderate to severe motion: when combined with a robust reference-based metric such as MoMRISim, it yields performance assessments that closely align with human evaluation. Because PMoC3D includes both instructed and naturally involuntary motion, we also perform an analysis of the involuntary cases. The results, presented in Appendix G.4.

### 4.3 Assessing simulated motion evaluation

Evaluation based on simulated motion is popular for its simplicity and since motion-corrupted real-world data is not required. However, such simulations may not reflect real-world motion artifacts sufficiently well. We find that evaluation based on simulated data overestimates the performance of reconstruction methods and can therefore be misleading.

This is easy to see from Figure 4, which contains a severely corrupted real scan from the PMoC3D dataset as well as a simulated motion with a similar level of motion corruption (as seen in the L1 reconstructions in the figure). It can be seen that the reconstructions of all considered algorithms are significantly better for simulated data compared to the reconstructions based on the real data. In addition, Figure 12 in Appendix G.2 compares five simulated cases of severe motion with the five most severe cases from real-world scans, showing results both with and without motion correction. The simulated artifacts were designed to be as severe as or more severe than those found in the real-world data. In all instances, reconstructions from the real-world data retain noticeable ringing artifacts. In contrast, the reconstructions of simulated data using motion correction appear consistently clean, with motion artifacts largely eliminated.

Our comparison confirms that, even at comparable levels of artifact severity, reconstructions from simulated motion consistently appear cleaner than those from real-world motion. This discrepancy indicates that evaluation approaches restricted to simulation therefore risk systematically overestimating algorithmic progress.

### 4.4 Assessing reference-free evaluation

Reference-free evaluation can test for the presence of artifacts, but cannot measure accuracy since no reference is available. We find that reference-free metrics systematically overestimate the performance of the deep learning based method.

To evaluate metric reliability, we examine the correlation between reference-free quality scores and PMAS. As illustrated in Figure 2, right panel, the reference-free VLM score(GPT-4o) exhibits a weak correlation with the human judge. Comprehensive results for all reference-free metrics are reported in Appendix G.1, where consistently low correlations with expert evaluation are observed. A notable failure case involves certain stacked U-net reconstructions, where the VLM score assigns high quality despite PMAS indicating substantial motion artifacts. This discrepancy suggests

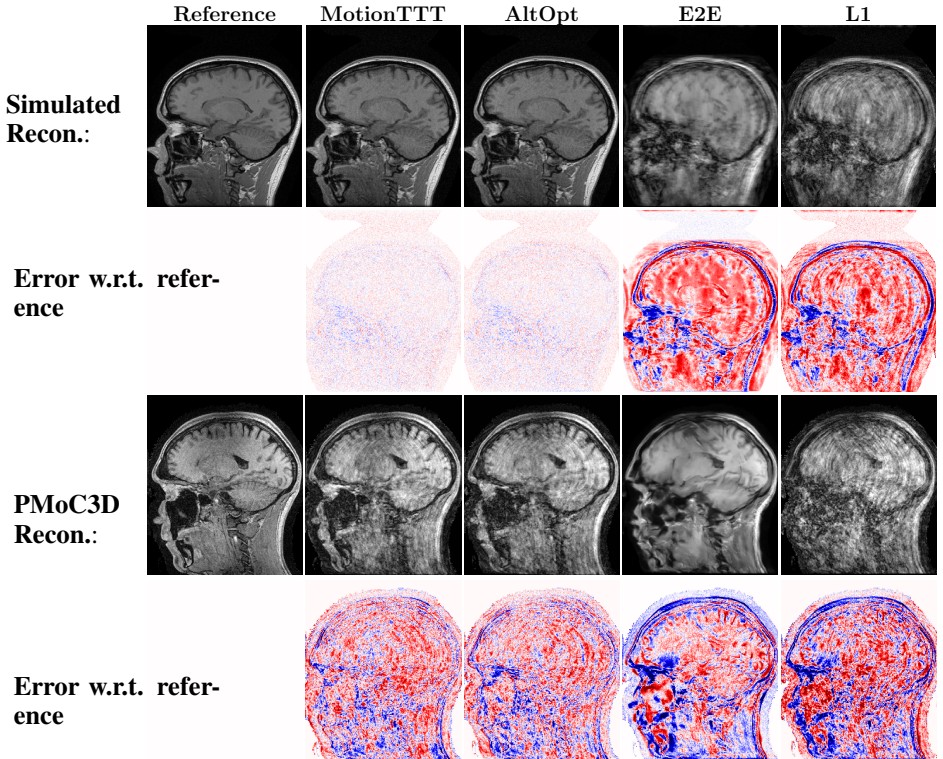

Figure 4: Comparison of baseline reconstructions on scans with artifacts from both simulation and the real-world PMOC3D dataset. The first column displays the motion-free reference. Subsequent columns show the results from MotionTTT, AltOpt, E2E Stacked U-Net, and L1 minimization without motion correction. The error maps, shown below each reconstruction, visualize the difference relative to the reference.

that reference-free metrics do not reliably align with human perception of artifact severity. In Appendix G.3, we provide an example where a reference-free metric fails to reflect the quality of a reconstruction. Although the stacked U-net rated favorably by the metric, the corresponding difference image reveals substantial loss of anatomical detail. Reference-free evaluation is prone to bias, particularly for end-to-end deep learning models that suppress artifacts through oversmoothing.

## 5 CONCLUSION

Reliable evaluation of motion correction algorithms in 3D MRI is fundamentally difficult due to the absence of good ground truth in real-world data. This paper presents a comprehensive assessment of three evaluation approaches: real-world evaluation with a reference scan, based on simulated motion, and reference-free assessment.

To enable real-world evaluation, we introduced the PMoC3D dataset consisting of paired motion-free and motion-corrupted scans. We find that real-world evaluation is well correlated with human judgment of reconstruction and is thus relatively reliable. However, for very mild motion, baseline reconstruction methods can produce better results than the motion-free reference, which potentially compromises the validity of reference-based evaluation in the mild-motion regime. Evaluation based on simulated motion can be misleading because simulated motion fails to capture the full complexity of real-world motion and tends to overestimate performance. However, evaluation based on simulated motion can still be useful for relative comparisons.

Reference-free evaluation can be very biased towards certain reconstructions and is not reliable, as expected.

**Ethical statement.** The local institutional review board approved the study (study number 2024-365-S-NP) in accordance with the ethical standards of the institutional and/or national research committee and with the 1964 Helsinki Declaration and its later amendments or comparable ethical standards. Prior informed consent was obtained from all individual participants.

**Reproducibility Statement.** All data, code, and reconstructed volumes associated with this work are publicly available. Acquisition details for the PMoC3D dataset are provided in Section 2 and Appendix B; the dataset and the reconstructed volumes are hosted at `https://huggingface.co/datasets/mli-lab/PMoC3D`. Training and inference configurations for the baseline reconstructions are detailed in Appendix D, and the corresponding code to reproduce these baselines is available at `https://github.com/MLI-lab/MRI_MotionTTT`. Our evaluation methods are described in Section 3.4, Appendix E, and Appendix F, with the evaluation code accessible at `https://github.com/MLI-lab/PMoC3D/tree/main`.

**Acknowledgements.** This work was supported by the Deutsche Forschungsgemeinschaft (DFG, German Research Foundation) - 456465471, 517586365 and the German Federal Ministry of Education and Research, and the Bavarian State Ministry for Science and the Arts.

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

## A  THE USE OF LARGE LANGUAGE MODELS (LLMS)

We utilized LLMs in the following two approaches during this research:

- For Writing Assistance: LLMs were used to refine the grammar and improve the clarity and readability of this manuscript.

- For Evaluation: As a core component of our methodology, vision-language models (VLMs) were employed to evaluate motion artifacts. This process is detailed in Section 3.4 and Appendix F.

## B  PMoC3D ACQUISITION DETAILS

In Table 2 we provide a list of all relevant sequence parameters used for the acquisition of our PMoC3D dataset.

Table 2: Sequence parameters of the PMoC3D dataset.

| Parameter | Value |
|---|---|
| Sequence | 3D T1-TFE |
| Sampling | Cartesian |
| Flip angle (deg) | 8 |
| TR (ms) | 6.7 |
| TE (ms) | 3.0 (shortest) |
| TFE prepulse / delay (ms) | non-selective invert / 1060 ms |
| Min. TI delay (ms) | 707 |
| TFE factor | 204 |
| TFE shots | 52 |
| TFE dur. shot / acq (ms) | 1742 / 1347 |
| Shot interval (ms) | 3000 |
| TFE prepulse delay (ms) | 1060 |
| Under-sampling factor | 4.94 |
| Half-scan factor Y / Z | 1 / 0.85 |
| Number of auto-calibration lines | 37 |
| Profile order | random |
| Field of view (FH x AP x RL, mm) | 256 x 221 x 170 |
| Acquisition matrix | 256 x 221 |
| Fold-over direction | AP |
| Fold-over suppression | no |
| Fat shift direction | F |
| Water-fat shift (pixels) | 1.6 |
| Saturation slabs | no |

Figure 5 illustrates an example from the sampled dataset, including the image volume, its corresponding k-space representation, and the undersampling mask pattern applied along 2 phase encoding directions.

## C  PERCEIVED MOTION ARTIFACT SCORE DETAILS

In order to evaluate the severity of motion artifacts in the L1 reconstructions, we first shuffle the reconstructions and conceal their labels. Then, two PhD students with expertise in machine learning and MRI reconstruction performed pairwise comparisons between the reconstructions of the 24 motion-corrupted scans. If both evaluators agree that reconstruction A has more severe artifacts than B then we assign rate $p(A > B) = 1$. If one evaluator judges A to be better and the other finds a similar level, then we assign $p(A > B) = 0.75$. If both evaluators find a similar severity level, or one finds one better and the other the other, we set $p(A > B) = 0.5$.

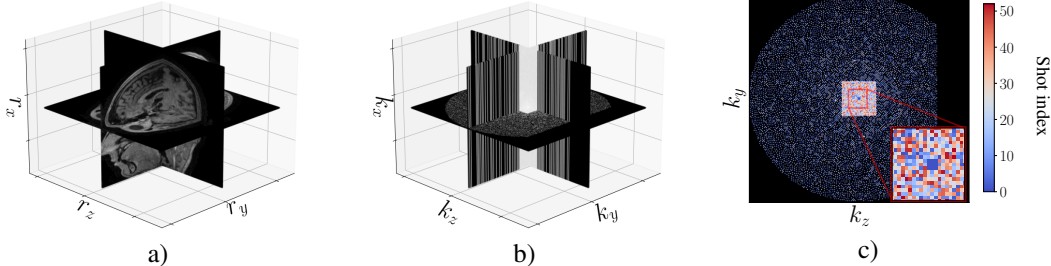

a)          b)          c)

Figure 5: Panel a): schematic visualization of the magnitude of a 3D volume; Panel b): the corresponding 3D k-space data. Panel c): the undersampling masks with the color coding illustrating an example of a random sampling trajectory indicating which lines along the readout dimension $k_z$ are sampled within the same out of 52 shots.

For the PMAS score described in Section 4.2, which is used to compare reconstruction quality across different baselines and scans, annotations were performed by a licensed medical doctor. The evaluation also follows a pairwise comparison protocol. For each pair of reconstructions, if the annotator judges that reconstruction A exhibits more severe artifacts than B, we assign a rate of $p(A > B) = 1$. If the two reconstructions are considered to have similar artifact severity, the win rate is set to $p(A > B) = 0.5$.

Based on those pairwise comparisons, we fit a Bradley–Terry model (Bradley & Terry, 1952), which assigns a latent parameter to each reconstruction. The difference in these parameters indicates which volume is considered to have more severe motion artifacts. We estimated these parameters by maximizing the likelihood function

$$\text{PMAS} = \arg\max_{\beta} \sum_{i \neq j} p(i > j) \log \left( \frac{\exp(\beta_i)}{\exp(\beta_i) + \exp(\beta_j)} \right), \tag{2}$$

using gradient descent with the Adam optimizer in PyTorch. In this formulation, each $\beta_i$ quantitatively represents the severity of motion artifacts for the corresponding volume; higher values indicate more severe artifacts, and these latent parameters serve as our perceived motion artifact score.

Table 3 presents each motion-corrupted scan's perceived motion artifact score in the PMoC3D dataset. Based on the perceived motion artifact score, we categorize the scans into three motion severity levels as follows:

- **Mild Motion**: S1_2, S7_3, S7_1, S3_3, S1_3, S4_2, S5_2, S2_3
- **Moderate and Severe Motion**: S3_2, S4_1, S7_2, S5_3, S6_2, S2_2 ,S8_2, S6_1, S4_3, S3_1, S5_1, S1_1, S8_1, S6_3, S8_3, S2_1

This classification facilitates a structured analysis of the reconstruction methods' performance across varying degrees of motion artifacts.

## D   Hyperparameter configurations and implementation details of baselines

In this Section, we provide further details regarding implementation and hyperparameter configurations used for reconstructing the PMoC3D dataset with classical L1-minimization (Lustig et al., 2008), alternating optimization similar to (Cordero-Grande et al., 2016), MotionTTT (Klug et al., 2024) and E2E Stacked U-net (Al-Masni et al., 2022) in Sections 2.2 and 4.1.

### D.1   L1-minimization

We perform L1-minimization with the mean-squared-error loss function and wavelet regularization. We use the Haar wavelet implementation of order one from the PyWavelets package (Lee et al.,

Table 3: Perceived motion artifact score (PMAS) for each motion-corrupted scan in the PMoC3D dataset. Higher scores indicate more severe motion artifacts.

| Scan ID | PMAS | Scan ID | PMAS | Scan ID | PMAS | Scan ID | PMAS |
|---------|------|---------|------|---------|------|---------|------|
| S2_1    | 2.417 | S4_3   | 1.552 | S4_1   | 0.808 | S7_1   | -0.156 |
| S8_3    | 2.230 | S6_1   | 1.405 | S3_2   | 0.771 | S7_3   | -0.231 |
| S6_3    | 2.197 | S8_2   | 1.400 | S2_3   | 0.485 | S1_2   | -0.443 |
| S8_1    | 2.189 | S2_2   | 1.102 | S5_2   | 0.356 |        |        |
| S1_1    | 1.828 | S6_2   | 1.041 | S4_2   | 0.346 |        |        |
| S5_1    | 1.782 | S5_3   | 0.867 | S1_3   | -0.008 |       |        |
| S3_1    | 1.673 | S7_2   | 0.813 | S3_3   | -0.057 |       |        |

2019) with PyTorch support through the `PyTorch Wavelet Toolbox` (Wolter et al., 2024). We run 40 steps with stochastic gradient descent, a learning rate of $10^8$ and regularization weight $\lambda = 3 \times 10^{-8}$.

## D.2 ALTERNATING OPTIMIZATION

To perform alternating optimization as described in Section 4.1 we run SGD with a learning rate of $10^8$ and regularization weight $\lambda = 3 \times 10^{-8}$ during the reconstruction steps and a learning rate of $5 \times 10^{-2}$ during the motion estimation step. In both steps the loss is the MSE between predicted and given measurement. The optimization process is capped at 500 iterations, but it terminates early if the difference between the losses of the first and second reconstruction step is less than 0.02.

After alternating optimization we perform L1-minimization from scratch based on the estimated motion parameters as described in Section D.1.

## D.3 MOTIONTTT

To conduct test-time-training motion estimation with MotionTTT (Klug et al., 2024) we use the model provided by the authors, which is pre-trained on the Calgary Campinas Brain MRI Dataset (Souza et al., 2018) for the task of 2D motion-free reconstruction from undersampled MRI.

As outlined in (Klug et al., 2024) the iterative motion estimation can be conducted in three phases, where during phase 2, motion states pertaining to shots that exhibit a large data consistency (DC) loss, can be split into several distinct motion states to estimate a more fine-grained motion trajectory during phase 2 and 3. This can improve the reconstruction quality compared to terminating MotionTTT after phase 1 as potentially less measurements have to be discarded during the DC loss thresholding before the final reconstruction.

For the PMoC3D dataset we observed no significant difference between the reconstruction quality of splitting corrupted shots during phase 2 or terminating the optimization after phase 1 and directly thresholding the corrupted shots from the reconstruction. The number of read-out lines that are saved from being thresholded during phase 2 lies in the range from 1-5% of the total number of lines, which appears to be too little to make a visual difference in the reconstruction.

Hence, for the results discussed here we reduce the computational costs of MotionTTT by running only phase 1, where one motion state (3 rotation and 3 translation parameters) is estimated per acquired shot. Specifically, we run 80 steps with an initial learning rate of 1.0 reduced by a factor of 2 at steps 50, 60 and 70. All other parameters are set as in (Klug et al., 2024).

For the final reconstruction we run L1-minimization as described in Section D.1 based on the estimated motion parameters, where shots with a DC loss larger than a threshold of 0.70 are excluded from the reconstruction.

## D.4 E2E STACKED U-NET

For the E2E Stacked U-net baseline results we adopt the network design from (Al-Masni et al., 2022), where we set the number of channels in first layer of both U-nets to 64 resulting in a total

of 15.9M network parameters. We used instance norm instead of batch norm in our network as we found it to give more stable results.

We train the model on the slices of 40 volumes from the Calgary Campinas Brain MRI Dataset (Souza et al., 2018). In every training step one 3D volume is loaded from which the fully sampled target volume is computed as well as motion-free and motion-corrupted undersampled input volumes. Then, 20/10 slices are selected randomly from the motion-free/motion-corrupted volumes in each plane $r_x \times r_y$, $r_x \times r_z$ and $r_y \times r_z$ together with the corresponding target slices resulting in a total of 90 input-target pairs per training step. Thus, with a batch size of 10 the network parameters are updated 9 times per training step and $40 * 9 = 360$ times per epoch. We train the model for 200 epochs with the SSIM loss and the Adam optimizer with a learning rate of $6 \times 10^{-4}$ which is decayed twice by a factor of 10 at epochs 130 and 170.

We use twice as many slices from the motion-free input to ensure that the network can achieve high quality reconstructions in the absence of motion. We generate the motion-corrupted volumes based on the inter-shot motion simulation model from (Klug et al., 2024), where we focus on very mild motion with either one or two motion events. During a motion event 1-6 randomly selected motion parameters are set to a value drawn uniformly from either $[-1, 1]$ degrees/mm or $[-2, 2]$ degrees/mm simulating subject movement in between the recording of two shots. We focus the model training on mild motion as for more severe artifacts image details are occluded and thus irreversible lost for reconstruction with an end-to-end approach. Nevertheless, we note that the motion correction capability of an end-to-end model is specific to the type of motion simulated during training and hence a more sophisticated motion simulation could benefit the model's performance in the regime of mild motion.

## E  IMPLEMENTATION OF MoMRISIM

We propose a motion MRI similarity(MoMRISim) as a learned perceptual metric to quantify motion-artifact severity in 3D MRI. The basic idea of the MoMRISim is described in Section 3.4. We are going to describe the implementation details on this section.

We train on 40 brain volumes from the Calgary Campinas (CC-359) dataset. For each training volume we apply rigid-body motion corruption to the k-space under 7 randomly sampled severity levels, where severity definitions-ranging from mild to severe in terms of number of motion events and rotation/translation perturbation magnitudes, with definitions following (Klug et al., 2024). And reconstruct each corrupted k-space without motion correction by both an L1-wavelet reconstruction and a 2D U-Net. This ensures that MoMRISim observes artifact patterns from both classical compressed sensing and deep learning pipelines.

Training samples are triplets $\langle \text{Ref}, C_1, C_2 \rangle$, where Ref denotes the L1-wavelet reconstruction of the original motion-free volume, and $C_1$, $C_2$ are reconstructions of the same slice under two distinct motion severities. For example, if the severity of $C_1$ lower than that of $C_2$, then Ref should be closer to $C_1$. An example of the triplet is shown in Figure 6.

In each epoch, we construct triplet by enumerating all pairs of reconstructions at two distinct severity levels among the seven corrupted volumes. And then randomly select one of the three anatomical planes (axial, coronal, sagittal), sample ten slices per pair, normalize each slice by its 99.9th-percentile intensity, and discard any background-only slices. This yields approximately 7 000 triplets per epoch.

We adapted the same training way as the DreamSim(Fu et al., 2023) to fine-tune the DINO-vitb16(Caron et al., 2021) visual encoder augmented with LoRA adapters by minimizing a hinge-ranking loss. The model was optimized with AdamW (learning rate $3.0 \times 10^{-5}$), LoRA rank 4, and a batch size of 8, over 40 epochs. Training was conducted on NVIDIA RTX A6000 GPUs and completed in approximately 90 minutes with 4 workers. The final model achieved a triplet-ranking accuracy of 0.933 on the training set.

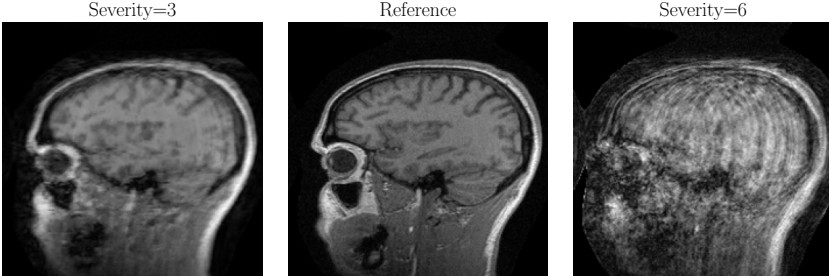

Figure 6: Example triplet from the MoMRISim training dataset. The left image is corrupted by motion (severity level 3) and reconstructed using a 2D U-Net. The right image is corrupted by more severe motion (level 6) and reconstructed with an L1 reconstruction without motion correction. The center image is a motion-free reconstruction using an L1 reconstruction.

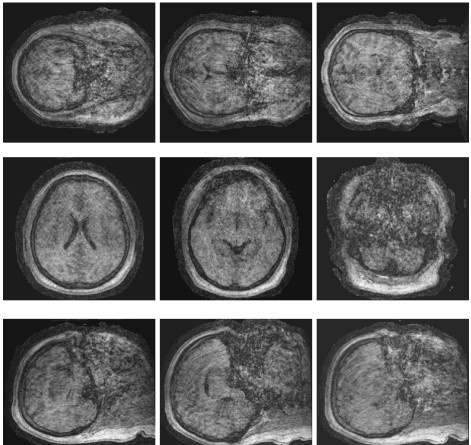

Figure 7: An example input for VLM evaluation, where three slices from sagittal, coronal, and axial orientations are arranged in a 3x3 grid for assessment.

# F    IMPLEMENTATION OF VLM SCORE

Vision-language models (VLMs) have found extensive applications across various domains. In this study, we employ multiple VLMs, each requiring a specific input format. For GPT-4o (OpenAI, 2024b;a) and Qwen2.5-VL-Max (Bai et al., 2025), we arrange three slice images from three different views into a 3x3 grid as their input. Figure 7 provides an example of this setup. Since Med3DVLM (Xin et al., 2025) and M3D-LaMed (Bai et al., 2024) support 3D input, we resize the volume to the desired size for each model's input.

We evaluated motion artifacts by generating five independent responses for each reconstruction using a temperature setting of 0.5. Each response is categorized into one of four predefined levels: No Motion, Mild, Moderate, or Severe, denoted as scores 0, 1, 2, and 3 respectively. The final evaluation score for each instance is calculated as the average of these five categorizations. The prompt we used is as follows:

```
**Task:**
Evaluate the severity of motion artifacts in the provided MRI image
    using a structured and systematic analysis.
---
### **Evaluation Criteria for MRI Image**
- **No Motion Artifact:** No visible motion artifacts; excellent
    diagnostic quality, and minor reconstruction noise is acceptable
    .
- **Mild:** The majority of brain details are clearly visible, with
     only minor artifacts that do not obscure diagnostic structures;
     minimal diagnostic impact, and minor reconstruction noise is
    acceptable.
- **Moderate:** Noticeable artifacts that partially obscure
    critical diagnostic regions; artifacts significantly impact
    diagnostic interpretation.
- **Severe:** Brain structures are predominantly obscured by
    artifacts, with only the general shape discernible; diagnosis is
     extremely challenging or impossible.
### **Output Template**
**Analyze Brain Structure Visibility**
    - Does the image look very smooth, potentially losing
        significant detail? *(Important for scoring!)*
    - Are all major brain details visible (gyri, sulci, ventricles)?
    - Do motion artifacts blur or distort critical brain details?
    - Are there regions where brain details are completely lost?
**Assess Artifact Types and Locations**
    - Check for ringing effects (where, how severe).
    - Identify other motion artifacts (streaking, ghosting) and note
        their severity.
**Oversmooth Assessment**
    - Does the image look very smooth (like a very high-quality
        image)?
    - Are there areas with smooth distortions?
    - If yes, do you think the image has an oversmoothing problem?
- The primary MRI image shows **[overall assessment]** motion
    artifacts. The final precise motion artifact level is: [No
    Motion/Mild/Moderate/Severe]
If the severity level is No Motion/Mild: Re-examine the image. Are
    all details truly clear? If any structures appear compromised,
    consider increasing the severity level.
---
### **Conclusion**
- After rethinking, the primary MRI image shows **[overall
    assessment]** motion artifacts, and the details are **. Given
    these factors, the final precise motion artifact level is:
Severity Level: [No Motion/Mild/Moderate/Severe]
```

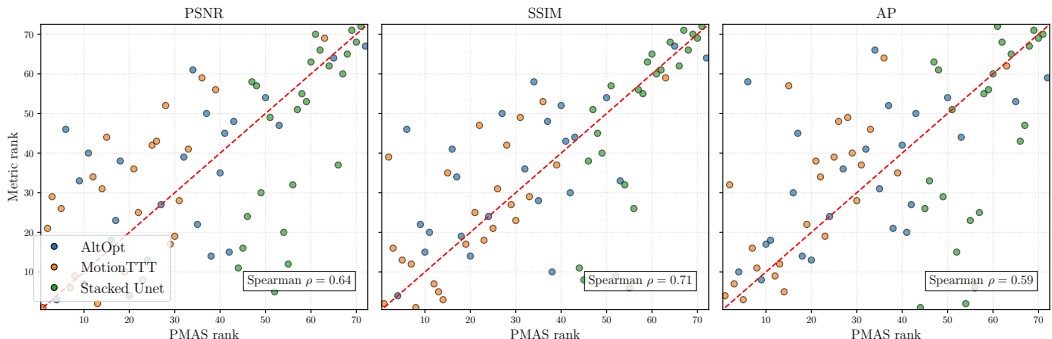

Figure 8: Rank comparison of PSNR, SSIM, and AP with the perceived motion artifact score. All of them show moderate correlation with human judgment.

## G  ADDITIONAL EXPERIMENT RESULTS

### G.1  CORRELATION ANALYSIS BETWEEN PMAS AND METRICS ON PMOC3D RECONSTRUCTIONS

This section presents the correlation results between the Perceived Motion Artifact Score (PMAS) and various image quality metrics computed on the PMoC3D reconstructions. The correlations are the figures for all mild, moderate, and severe situation.

Figure 8 shows the correlation between PMAS and traditional pixel-wise metrics. While these metrics generally reflect the expected trend of increasing degradation with higher motion artifacts, their correlation with PMAS is only moderate. This suggests that pixel-wise metrics have limited sensitivity to perceptual quality differences caused by motion corruption and may not be fully reliable for evaluating motion-degraded 3D MRI.

Figure 9 presents the correlation between PMAS and feature-based metrics. Overall, these metrics show a strong correlation with PMAS. However, DISTS demonstrates a notable failure mode when evaluating reconstructions from the stacked U-net, consistently assigning them abnormally low scores. In contrast, both DreamSim and MoMRISim exhibit more stable behavior and higher alignment with PMAS. Among them, MoMRISim achieves the highest correlation with PMAS, indicating its robustness in capturing motion artifacts.

Figure 10 and Figure 11 presents the correlation between PMAS and reference-free metrics, which overall show poor alignment with human judgment. Among the evaluated metrics, the VLM scores evaluated by GPT-4o get the highest correlation. And the VLM score(GPT-4o) also demonstrates significantly better alignment with PMAS than TG and AES. These results suggest that while reference-free metrics are generally less reliable for assessing motion artifacts, the VLM score(GPT-4o) may offer a alternative when references are unavailable.

### G.2  COMPARISON BETWEEN SIMULATED AND REAL-WORLD EVALUATION UNDER SEVERE MOTION

Figure 12 shows five sets of reconstructions with and without motion correction under severe motion artifacts, comparing PMoC3D and simulated motion cases. The first two columns show L1 reconstructions without motion correction, which reflect the raw severity of motion artifacts. Both real and simulated scans display strong artifacts. Notably, the real-world L1 reconstructions still preserve some anatomical details, while the simulated counterparts often obscure brain structures entirely-indicating that the simulated artifact severity is comparable to or even greater than that of real-world scans.

The last two columns present the corresponding MotionTTT reconstructions. In all cases, real-world data retains noticeable ringing artifacts, with the fifth row showing particularly obvious artifacts. In

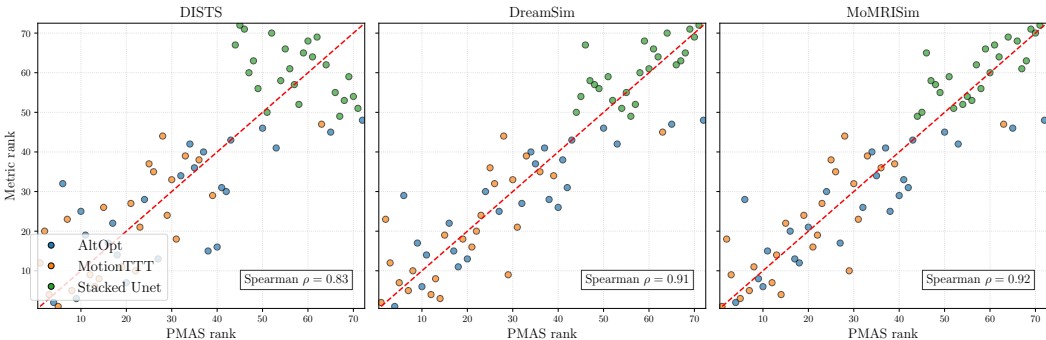

Figure 9: Rank comparison of DISTS, DreamSim, and MoMRISim with the perceived motion artifact score. All of them show high correlation with human judgment, while the MoMRISim shows the highest correlation.

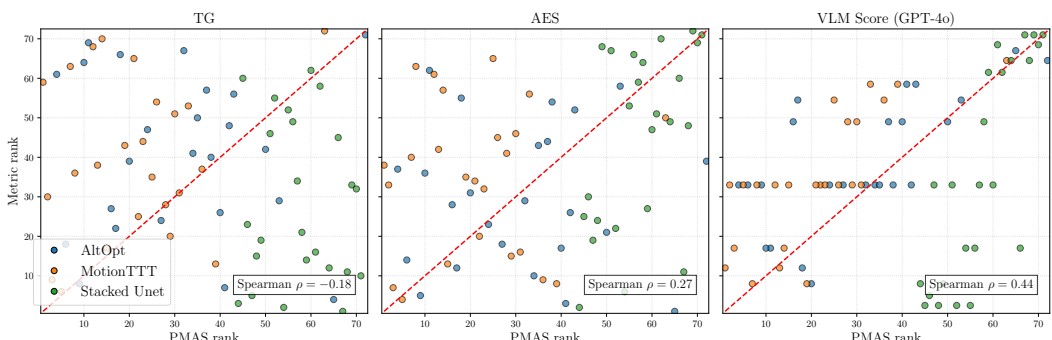

Figure 10: Rank comparison of TG, AES, and VLM Score(GPT-4o) with the perceived motion artifact score. All of them show poor correlation with human judgment.

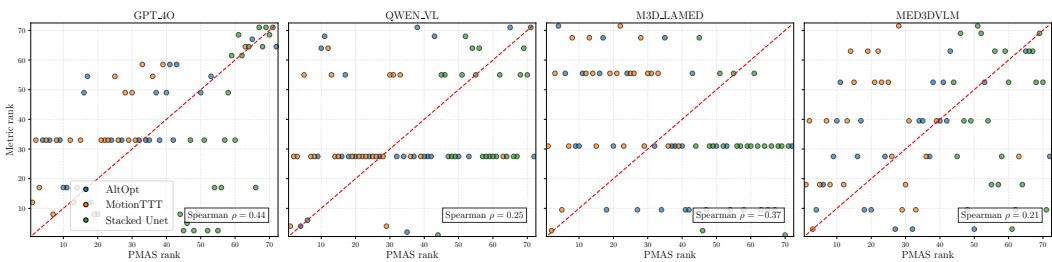

Figure 11: Rank comparison of VLM scores evaluated by GPT-4o, Qwen-VL-Max, Med3DVLM, and M3D-LaMed with the perceived motion artifact score. All of them show poor correlation with human judgment.

Real(L1 noMoCo)     Simulate(L1 noMoCo)     Real(MotionTTT)     Simulate(MotionTTT)

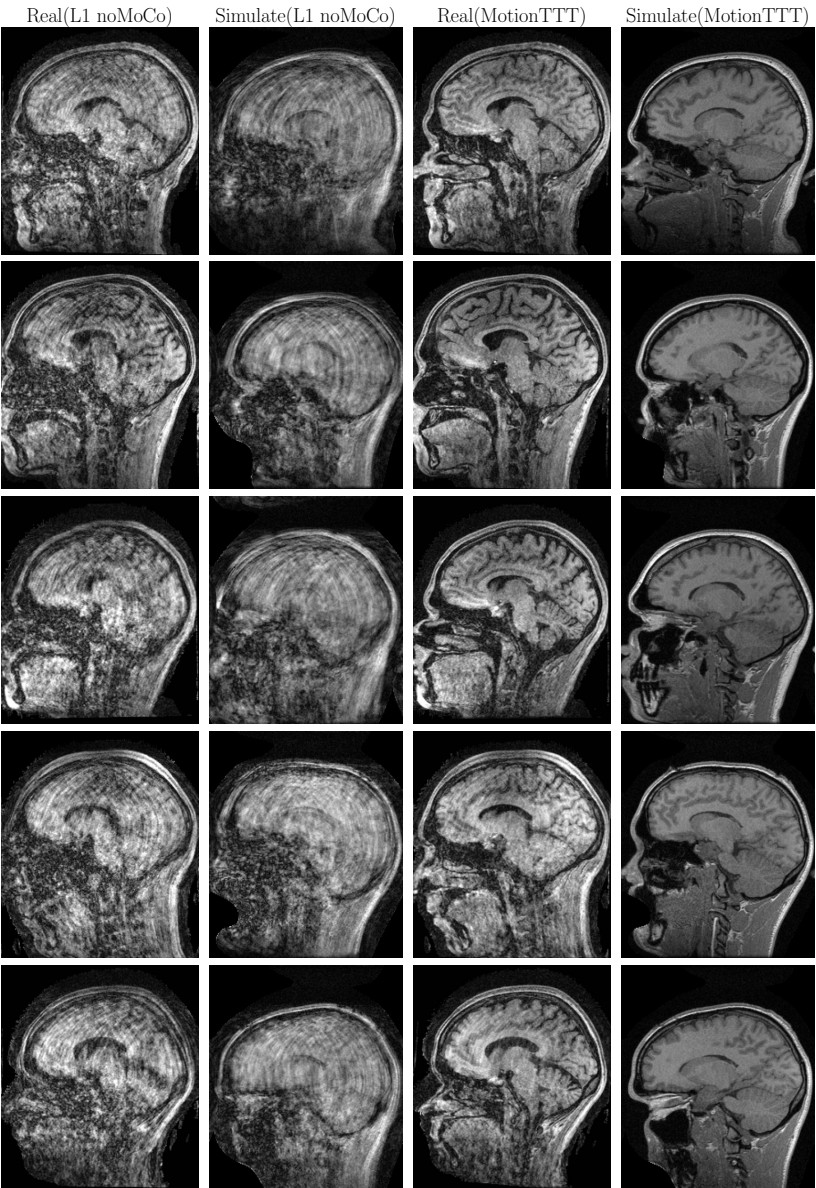

Figure 12: Reconstructions from the PMoC3D (real) and simulated datasets under two reconstruction methods. The first two columns show L1 reconstructions without motion correction, where simulated volumes exhibit motion artifacts of similar or greater severity compared to real-world scans. The last two columns display MotionTTT reconstructions: while real scans retain visible motion artifacts, simulated volumes are consistently corrected with minor residual artifacts.

contrast, the simulated MotionTTT reconstructions appear consistently clean, with motion artifacts largely eliminated.

Given that the simulated artifacts are at least as severe as those in real-world scans, the significantly better reconstruction quality further confirms that simulation-based evaluation can lead to a systematic overestimation of reconstruction performance in practical settings.

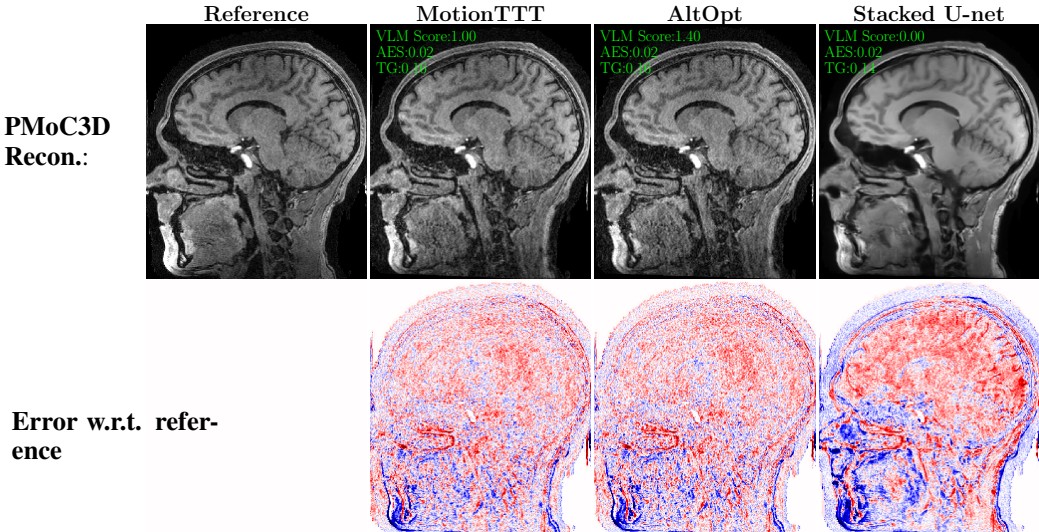

Figure 13: Baseline reconstructions of scan S4_2, with the difference images and the calculated reference-free scores.

### G.3 FAILURE EXAMPLE OF REFERENCE-FREE EVALUATION

In this section, we present an additional failure case of reference-free evaluation. Figure 13 shows reconstructions from different baselines along with their corresponding error maps.

As illustrated, the stacked U-Net reconstruction exhibits substantial loss of anatomical detail, as clearly visible in the error image. This result is qualitatively worse than those produced by MotionTTT and AltOpt. However, due to its visually smooth appearance, the stacked U-Net receives comparable scores from TG and AES, despite its degraded quality.

Even the VLM score(GPT-4o) while generally better aligned with human judgment fails in this case, assigning a near-perfect score to the stacked U-Net reconstruction. This example underscores a key limitation of reference-free metrics: they can be misled by superficially clean outputs that actually lack critical structural fidelity.

### G.4 RESULTS OF INVOLUNTARY SCANS

Our dataset also includes a small number of involuntary-motion scans. One of these exhibits significant anatomical misalignment with its reference scan and is therefore excluded from the analysis. We examine the two involuntary scans in Figure 14, which show their baseline reconstructions and corresponding error maps relative to the references. The observed artifacts primarily consist of faint ringing, similar to that seen in voluntary mild-motion cases. After applying reconstruction methods such as MotionTTT, the resulting image quality is comparable to that of the corresponding reference-free reconstruction, and no unexpected failure modes are observed.

We also analyzed the metric scores of the reconstructions from the two involuntary scans. Starting from the 250 expert comparisons used for voluntary-motion volumes, we randomly added 63 additional pairwise comparisons, including one involving the involuntary reconstructions, and refit the Bradley–Terry model as described in Section 2.2. We further computed the MoMRISim values for the same involuntary reconstructions. The resulting rank–rank relationship is shown in Figure 15. The involuntary scans fall within the normal variability of voluntary motion cases and do not appear as outliers. This figure indicates that the available involuntary scans do not exhibit abnormal behavior relative to the voluntary cases.

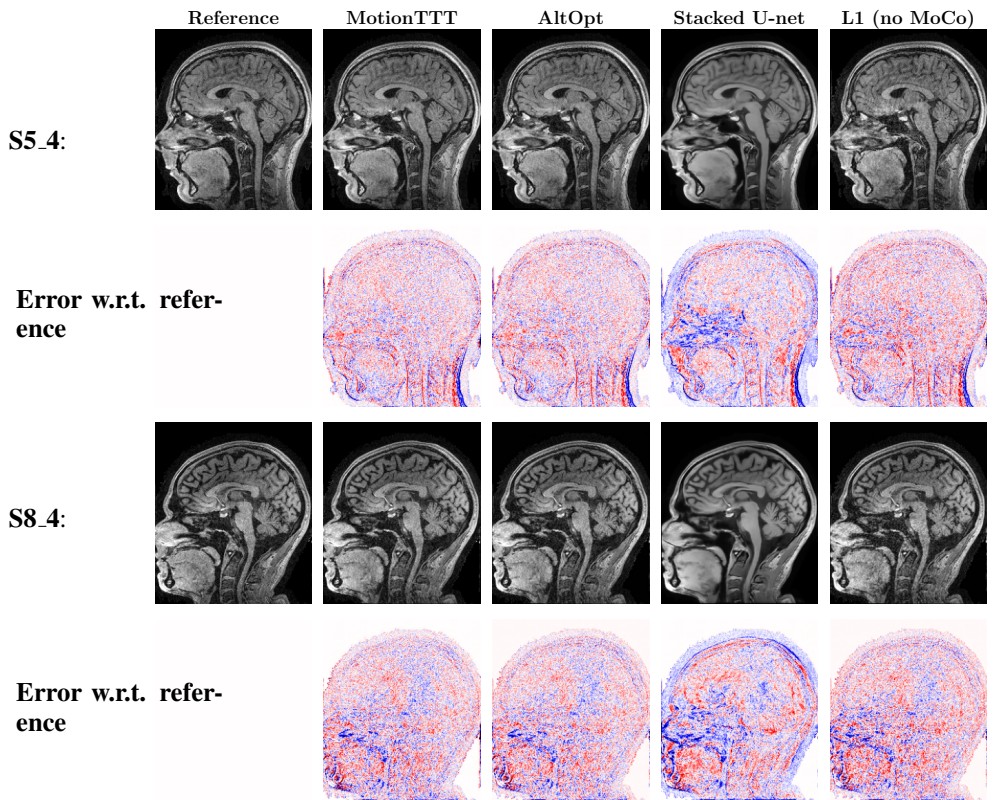

Figure 14: Baseline reconstructions of involuntary motion scans S5_4 and S8_4, with the difference images and the calculated reference-free scores.

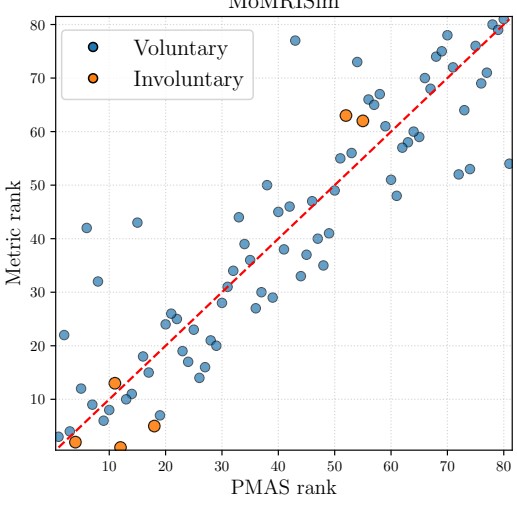

Figure 15: Rank comparison of MoMRISim with the perceived motion artifact score for both voluntary motion and involuntary motion scans.

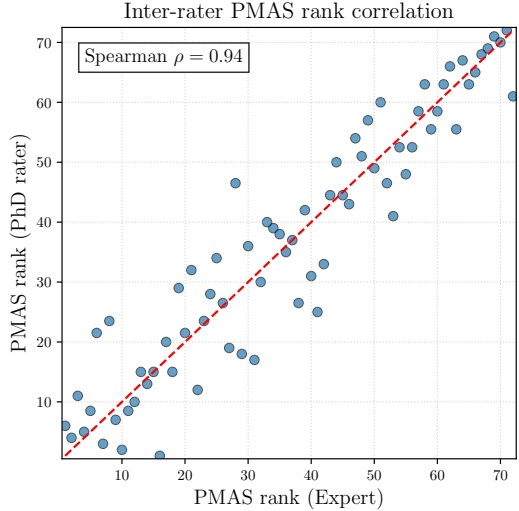

Figure 16: Rank comparison of the perceived motion artifact score from 2 raters.

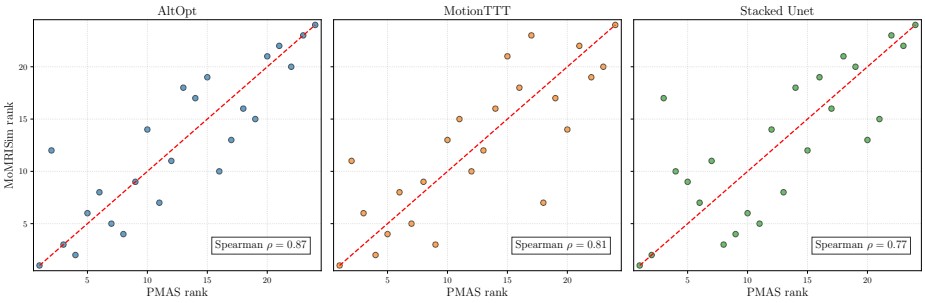

Figure 17: Rank comparison between MoMRISim and the perceived motion artifact score within different reconstruction methods.

## G.5 INTER-RATER ANALYSIS

To assess the reliability of using a single rater for PMAS, we asked a second evaluator (a PhD student with extensive experience in motion-corrupted MRI reconstruction) to independently score all 72 reconstructed volumes. To reduce evaluation noise, this rater performed a full set of pairwise comparisons, i.e., 2,556 comparisons covering all reconstruction pairs, and we then fitted a Bradley–Terry model to obtain PMAS values for the 72 reconstructions. As shown in Figure 17, the resulting ranking exhibits a very strong Spearman correlation of 0.94 with the medical doctor's PMAS. This high agreement demonstrates that PMAS is reproducible and not dependent on a single rater's subjective preference.

## G.6 WITHIN-METHOD CORRELATION ANALYSIS

Because each scan is reconstructed with multiple methods, a single volume contributes multiple points in Figure 2. To verify that the observed correlation is not driven by differences between reconstruction pipelines, we also computed correlations within each method separately. As shown in Figure 17, using the 24 scans reconstructed by each method, the correlations between MoMRISim and PMAS are 0.87 for AltOpt, 0.81 for MotionTTT, and 0.77 for the Stacked U-Net. The consistently high within-method correlations indicate that MoMRISim reflects motion severity rather than method-specific variation.

