# OpenReview forum: "Reliable Evaluation of MRI Motion Correction: Dataset and Insights"
_ICLR.cc/2026/Conference — ICLR 2026 Poster_

### Official Review · Reviewer_YT8x · 2025-10-24

**Soundness:** 2
**Presentation:** 2
**Contribution:** 2
**Rating:** 4
**Confidence:** 4

**Summary:**

This paper attempts to address the evaluation problem for MRI motion correction algorithms. It contributes a small-scale (n=8) paired dataset (PMOC3D) and a new metric (MoMRISim). The authors use this to argue that simulation-based evaluation is misleading and reference-free metrics are biased.

**Strengths:**

The paper focuses on an important problem of MRI motion correction about reliable evaluation. It provides some experimental evidence pointing out the pitfalls of simulation and reference-free evaluations. The MoMRISim metric shows a high correlation with a single rater on this specific dataset.

**Weaknesses:**

1. The core contribution (a dataset and a metric) is weak. First, a dataset of n=8 subjects is too small to serve as a meaningful or generalizable benchmark. Second, the concept of paired scans is not novel in the medical imaging domain (e.g., in infant datasets like BCP). The work feels more like a domain-specific dataset paper than a methodological contribution suitable for ICLR.
2. The paper's entire argument is self-contradictory. The authors' central thesis is that simulated data is "unreliable". Yet, they use this "unreliable" simulated data to train their proposed metric MoMRISim.
3. The benchmark's clinical relevance is questionable. All quantitative conclusions are derived exclusively from instructed voluntary motion. The paper fails to provide any analysis of the involuntary motion samples mentioned in the dataset, thus lacking evidence that instructed motion is a valid proxy for the true clinical challenge.
4. The paper's core claims suffer from low statistical power. The conclusions are drawn from a small sample size of only eight subjects. Critically, the gold-standard PMAS used to validate the MoMRISim metric was established by a single licensed medical doctor, with no inter-rater reliability analysis. This design means the headline claim (MoMRISim's $\rho=0.92$ correlation) lacks statistical significance. The method's applicability is also severely limited, as the authors concede their reference-based approach is invalid in the mild-motion regime.

**Questions:**

1. Please discuss the core contradiction (Weakness #2): Why should a metric trained on "unreliable" simulated data be trusted to evaluate real data?
2. Please provide an analysis to justify why instructed motion (used in the paper) is a valid proxy for involuntary motion.
3. Please comment on how the low statistical power (n=8 subjects, 1 rater) affects the confidence in the claim that MoMRISim is the best metric.

---

> ### Author Response · Authors · 2025-11-24
>
> We thank the reviewer for the feedback. We address each of the concerns raised on a point-by-point basis below.
> - **Weakness 1 Core Contribution/dataset size**
> The contribution of this paper is to systematically advance benchmarking for motion correction, which is an emerging topic in deep learning for imaging. Towards advancing benchmarking, we contribute the following:
>
>     - A dataset, PMoC3D. While we agree that he number of subjects appears small, it is consistent with common practice in imaging and sufficient for benchmarking. Many related studies rely on only 3-10 subjects or volumes for both training and evaluation. Our dataset comprises 27 motion-corrupted volumes for evaluation, which is sufficient for benchmarking image reconstruction and motion correction methods. Public in vivo datasets are rare, especially those that release full raw k-space data.
>     - Methodological contribution: Beyond the dataset itself, the central methodological contribution of this paper is a systematic study of three major evaluation methods (paired, simulated, and reference-free) for 3D MRI motion correction. This has not been done before and provides important insight into benchmarking: We have concluded that the paired real-world evaluation, although not perfect, is the most reliable. And the method, such as simulated motion, which is widely used in many research studies, systematically exaggerates algorithm performance. This analysis is methodological and provides evidence on how future work should evaluate 3D motion-correction methods.
> - **Response to Weakness 2 and Question 1**
> There is no contradiction, because simulation serves two different purposes, and its "reliability" is context-dependent:
>     - For assessing evaluation - unreliable: For assessing evaluation,  simulation is unreliable because it provides an overly optimistic estimate of the final reconstruction quality. Simulated motion often lacks the complexity of real motion, leading to artificially clean reconstructions.
>     - For metric training - effective: For training MoMRISim, we need a large set of images with known, ordered levels of motion severity. Simulation is perfectly suited for this because it provides us with unlimited, perfectly labeled training triplets. And the reconstructions used for generating these training triplets do not involve any motion-correction algorithms. The metric's goal is to learn a robust feature-space representation of motion artifact severity that generalizes well to real data, as demonstrated by its high correlation with human experts on the real-world PMoC3D set.
> - **Response to Weakness 3 and Question 2**
> The physics of MRI artifact formation depends on the motion itself, not whether it was voluntary or involuntary. Identical movements produce identical k-space inconsistencies regardless of intent, and our motion instructions capture the most common real-world head movements. Capturing matched involuntary motion with ground truth is extremely challenging; instructed motion offers a controlled and scalable alternative.
>
>     To assess how involuntary motion behaves in our dataset, we performed an additional analysis using involuntary scans in Appendix G.3. The artifact patterns (e.g., ringing) in these scans are qualitatively similar to those in voluntary mild-motion cases. After applying reconstruction methods such as MotionTTT, no unexpected failure modes are observed. Moreover, their PMAS and MoMRISim ranks fall within the normal variability of voluntary scans and do not appear as outliers. While the number of involuntary cases is limited, this provides initial evidence that instructed motion is a reasonable proxy for involuntary cases. Therefore, the evaluation benchmark is clinically relevant.
>
> - **Response to Weakness 3 and Question 4**
>     - Rater reliability: To evaluate the stability of the expert’s ranking, we asked a second evaluator (a PhD student with extensive experience in motion-corrupted MRI reconstruction) to score all 72 reconstructed volumes. To minimize noise, this evaluator performed full pairwise comparisons(2556 comparisons). This evaluation exhibited a strong Spearman correlation ($\rho = 0.94$) with the medical doctor’s ranking. This demonstrates that PMAS is reproducible and not dependent on a single rater’s subjective preference. A detailed analysis is updated in Appendix G.4.
>
>     - Score reliability: We evaluate effectiveness using 8 subjects, each with 3 motion-corrupted scans reconstructed using 3 different methods, resulting in 72 independent data points. Although the number of subjects is limited, this sample offers sufficient variability in motion type, severity, and reconstruction method to support meaningful correlation and ranking analyses.
>
> We hope this addresses the reviewer's concerns, and if not, we're happy to clarify further.

---

### Official Review · Reviewer_PvMP · 2025-10-29

**Soundness:** 2
**Presentation:** 1
**Contribution:** 2
**Rating:** 2
**Confidence:** 4

**Summary:**

This paper presents two main contributions. The first is a curated dataset comprising T1-weighted brain MR images from eight subjects. For each subject, scans were acquired with four distinct levels of induced motion artefacts, including a reference scan presumed to be motion-free. Each image is accompanied by a "perceived motion artefact score."

The second contribution is the proposal of two novel metrics for quantifying motion artefact severity. The first, MoMRISim, is a reference-based metric. It is an adaptation of the DreamSim approach (Fu et al., 2023) that is specifically trained on data with simulated motion severity levels instead of manually annotated images. The second is a reference-free metric, termed the VLM score, which is derived by prompting a Vision-Language Model.

The authors conduct an analysis to study and compare the behaviour of these proposed metrics against existing methods using their newly introduced dataset. They investigate the correlation between visual ratings and both reference-based and reference-free metrics, as well as the variations in performance when using synthetic motion-corrupted MRIs compared to images with real motion.

**Strengths:**

- **Valuable dataset**: The creation of a dataset with real MRI scans acquired at different, controlled levels of motion artefact, including a motion-free reference for each subject, is a significant strength. Such data is difficult and costly to obtain, and it provides a valuable resource for developing and rigorously evaluating motion detection and correction algorithms.
- **Novel and relevant metrics**: The introduction of two new metrics—a tailored reference-based metric (MoMRISim) and a novel reference-free metric using a Vision-Language Model (VLM score)—addresses an important need in the field. The goal of developing metrics that align with human visual perception of artefact severity is highly relevant and practical for automated quality control.

**Weaknesses:**

- **Poor clarity and structure**: The paper's organisation significantly hinders comprehension. The introduction lacks a foundational explanation of motion artefacts, their impact, and existing detection/correction methods. Key details, such as the derivation of the "perceived motion artefact score" (Section 2.2), are insufficiently explained in the main text. Section 3, which covers evaluation approaches, confusingly mixes descriptions of existing and proposed solutions without clear distinction. The rationale for selecting the three specific motion reconstruction methods used for evaluation is not provided, making the experimental design appear arbitrary.
- **Unclear methodological contributions**: The technical description of the two proposed metrics is unclear. For the reference-based MoMRISim metric, it is not adequately explained how the adaptation of DreamSim differs from the original work beyond the use of simulated motion data for training. For the reference-free VLM score, the prompting strategy and the specific Vision-Language Model used are not described with sufficient detail to understand or reproduce the method.
- **Ambiguous evaluation**: The design and interpretation of the results are problematic. Figures like Figure 2 are unclear; it is not evident if each data point represents a unique image or if the same subject/image is represented multiple times for different reconstruction types. This raises a critical concern that the observed correlations may be driven by the differences between reconstruction methods rather than by the metrics' ability to assess motion severity within a consistent context. This potential confound undermines the validity of the conclusions.

**Questions:**

- **Figure 2 interpretation**: Could you clarify what each data point in Figure 2 represents? Specifically, does a single subject/image appear multiple times for different reconstruction types, and if so, how does this affect the correlation analysis?
- **Metric details**: For the VLM score, please specify the exact prompt and the Vision-Language Model used. For MoMRISim, what are the key differences in the training protocol compared to the original DreamSim?
- **Baseline selection**: What was the rationale for choosing the three specific motion reconstruction methods used in the evaluation?
- **Motion simulation parameters** (Section 3.2): How were the number of events and the amplitude of the motion simulation parameters chosen?

Typos :
- In legend for Fig.1, images S7_1 and S5_1 are mentioned, albeit not actually appearing in the figure.
- Page 7 line ~34: mention of ‘AltOpt’, but abbreviation not previously introduced in first element of bullet-point list in part 4.1
- Page 9 line ~482: missing d letter “the full complexity of real-world motion and tends to […]”

---

> ### Author Response · Authors · 2025-11-24
>
> We thank the reviewer for the feedback. We address each of the concerns raised on a point-by-point basis below.
> - **Weakness 1: clarity and structure of paper**
> We thank the reviewer for this detailed feedback on clarity. We have made the following revisions to improve the paper's clarity:
>     - We expanded the introduction and now provide more background on motion artifacts, impact, and existing correction methods, as requested.
>     - For space reasons, we initially detailed PMAS in the appendix. As requested, we added the detailed derivation of the PMAS, including the Bradley-Terry model, directly into Section 2.2.
>     - For Section 3, we have added an overview table and reorganized Section 3.4 to clearly distinguish existing metrics from our proposed metrics.
>     - We added our rationale for selecting the three specific reconstruction baselines in Section 4.1
> - **Response to Weakness 2 and Question 2**
>
>     Regarding the technical description of the proposed metrics: For the VLM score, all details including the prompts used were included in the original submission. As described in Section 3.4, we used GPT-4o, Qwen2.5-VL-Max, Med3DVLM, and M3D_LaMed. The prompt used for all VLMs is provided in Appendix F. Figure 2 also explicitly labels the corresponding VLM. All code for reproducing the VLM score is included in the supplementary material.
>
>     Regarding differences in the training protocol of DreamSim and MoMRISim: The training pipeline is identical to that of DreamSim, and the hyperparameter settings are provided in Appendix E. The key differences lie in the task definition and supervision signal. DreamSim learns general perceptual similarity on natural images using large-scale human annotations. In contrast, MoMRISim learns to rank motion severity in MRI without any manual labels. We construct triplets with a perfectly known relative ordering, and train the model to learn a feature space specifically tailored to motion-artifact severity. This constitutes a task-level adaptation of DreamSim’s framework. We have updated Section 3.4 with a better clarification.
> - **Response to Weakness 3 and Question 1**
> For Figure 2, we have 8 subjects, each with 3 motion-corrupted volumes, resulting in 24 distinct volumes for evaluation. Each corrupted volume is reconstructed using three different motion-correction methods, so each original volume appears three times, once per reconstruction method. This produces 72 points in total.
>
>     Because each underlying scan contributes multiple points, the reviewer raised the concern that the correlation in Figure 2 might be driven by systematic differences between reconstruction methods rather than by the metric’s ability to assess motion severity. To address this, we recomputed the correlations within a single reconstruction method.
>     We compute correlations between MoMRISim and PMAS within each reconstruction method (24 scans per method), yielding 0.87 (AltOpt),  0.81 (MotionTTT), and 0.77 (Stacked U-Net). These consistently high correlations with PMAS demonstrate that MoMRISim’s alignment with expert PMAS is not driven by between-method shifts, but rather reflects its ability to accurately rank motion severity.
> - **Question3 Baseline selection**
> We selected these three methods because they represent three typical categories of commonly used 3D motion-correction algorithms in MRI:
>     - classical iterative optimization approach (AltOpt);
>     - hybrid method combining deep learning with physics-based reconstruction (MotionTTT);
>     - pure deep learning, end-to-end reconstruction model (Stacked U-Nets).
>
>     Together, these three baselines provide a representative and balanced coverage of the major methodological families in 3D MRI motion correction.
> - **Question 4 Motion simulation parameters**
> The motion simulation parameters were chosen to be similar to our real-world data. The number of events (one for mild, three for severe) was selected to match our PMoC3D acquisition, where subjects received up to three motion instructions.
> The amplitudes were then calibrated by manually inspecting the L1 reconstructions of the severe cases to ensure that the resulting simulated artifacts were visually similar to, or more severe than, those in our most challenging real-world scans.
>
> All noted typos have been corrected in the updated manuscript.
>
> We hope this addresses the reviewer's concerns, and if not, we're happy to clarify further.

---

> > ### Comment · Reviewer_PvMP · 2025-11-25
> >
> > I would like to thank the authors for their detailed responses and the clarifications provided, especially regarding the interpretation of correlation results and the rationale for the baseline selection. I appreciate that the authors addressed the individual within-method correlation values, although it would have been helpful to see the corresponding plots.
> >
> > Although these clarifications are useful and the dataset is clearly valuable, I still feel that the paper is not methodologically focused enough for ICLR. The main contribution remains the dataset, which would likely have greater impact at a specialised medical imaging venue.

---

### Official Review · Reviewer_ygbg · 2025-10-30

**Soundness:** 3
**Presentation:** 1
**Contribution:** 3
**Rating:** 6
**Confidence:** 5

**Summary:**

This work aims to construct a reliable and standardized evaluation pipeline for the MRI motion correction (MoCo) problem. The main contributions include three parts: 1) Collecting PMOC3D, a real-world MRI dataset; 2) Introducing MoMRISim, a feature-based MoCo metric; 3) Benchmarking several MoCo methods (including both traditional iterative and deep learning-based approaches) on simulated and real PMOC3D datasets. Overall, this work builds a comprehensive and reliable evaluation framework for the MRI MoCo task. It is interesting work.

**Strengths:**

- The collected PMOC3D dataset includes paired motion-free and motion-corrupted 3D k-space raw data, covering diverse motion types. Its construction and release can enable the development of more advanced deep learning–based MoCo methods, which is a critical contribution.
- The work explores many quantitative evaluation metrics for MoCo reconstructions, including reference-based metrics (e.g., pixel-level SSIM and PSNR, feature-level DreamSim) and reference-free metrics (e.g., AES, TG, and VIM scores based on LLMs). Moreover, a new feature-based metric, MoMRISim, is introduced. These studies on evaluation metrics may provide valuable tools for assessing MRI MoCo accuracy.
- Benchmarking several recent MoCo models on both simulated and real datasets provides useful insights that could guide future development of more advanced MoCo approaches.

**Weaknesses:**

- The PMOC3D dataset only includes 8 subjects. While I understand that MRI data collection is costly and time-consuming, this number is still relatively small. Therefore, the dataset may be more suitable as a test set rather than for training deep learning models.
- To my knowledge, the combination of deep learning models and physics-based iterative reconstruction is currently a mainstream paradigm for MRI MoCo [1][2][3]. These methods typically involve motion trajectory estimation. However, the proposed dataset does not provide the motion trajectories of the scanned subjects, which may limit its usability for evaluating many advanced MoCo methods.
- For building PMAS, two Ph.D. students performed the classification of motion types. I believe involving clinical experts in this process would improve the reliability of the annotations.
- Regarding the proposed MoMRISim metric, the paper lacks a sufficient intuitive explanation. As a reader unfamiliar with DreamSim, I found it difficult to understand the principle and motivation behind MoMRISim (Lines 270–286).

[1] Levac, Brett, et al. "Accelerated motion correction with deep generative diffusion models." Magnetic Resonance in Medicine 92.2 (2024): 853-868.

[2] Wu, Qing, et al. "Moner: Motion Correction in Undersampled Radial MRI with Unsupervised Neural Representation." The Thirteenth International Conference on Learning Representations.

[3] Singh, Nalini M., et al. "Data consistent deep rigid MRI motion correction." Medical imaging with deep learning. PMLR, 2024.

**Questions:**

See Weaknesses, please.

---

> ### Author Response · Authors · 2025-11-24
>
> We thank the reviewer for the feedback and the positive evaluation of our work. We address each of the concerns raised on a point-by-point basis below.
> - **Weakness 1, Limited Dataset Size**:
> We agree that the number of subjects is limited, but it is consistent with common practice in this research area. Many related studies [1][2][3] rely on only 3-10 subjects or volumes for both training and evaluation. Our dataset comprises 8 subjects and 27 motion-corrupted volumes for **evaluation**. As the reviewer mentions, this is sufficient for benchmarking image reconstruction and motion correction methods, but it is too little and also not meant for training. Public in vivo datasets are rare, especially those that release full raw k-space data. We believe our dataset adds meaningful value to the field because it provides this information.
> - **Weakness 2, Motion Trajectory**:
> We agree that additional true motion trajectories would be a valuable addition to our dataset. Our hardware setup does not include a motion-tracking system, but as a key enhancement to the dataset, we now provide a pseudo ground truth trajectory, estimated via a reference-guided optimization procedure.
>
>     We estimate a motion trajectory for each scan, using the MotionTTT trajectory (the best baseline in our experiments) as the initialization. These parameters are then refined by minimizing the mismatch between the motion-free reference reconstruction and the motion-compensated reconstruction. Given the motion-corrupted data $X$ and the reference-free volume $X_{\mathrm{ref}}$, we estimate the motion parameters by optimizing:
>
>     $\hat{m} = \arg\min_{m}
>     \frac{\left\lVert
>             X_{\mathrm{ref}} - U_{\text{net}}\!\left( T^{-1}(m)\, X \right)
>         \right\rVert_{1}}
>         {\left\lVert X_{\mathrm{ref}} \right\rVert_{1}}$
>
> where $T^{-1}(m)$ applies inverse motion to the corrupted data and $U_{\text{net}}(\cdot)$ reconstructs volumes from the motion-compensated data. This refinement begins with the baseline trajectory and applies gradient-based updates to further minimize residual artifacts. This can serve as a pseudo ground truth trajectory that we will include in the dataset for use in future research.
>
> - **Weakness 3, PMAS**:
> We appreciate the reviewer’s comment and apologize for the lack of clarity. The paper uses PMAS evaluations based on two different kinds of purposes:
>     - The PMAS used to assess the initial scan severity (Fig. 1) was conducted by two PhD students and was only used to categorize task difficulty.
>     - The PMAS used for evaluating reconstruction quality (Fig. 2), which supports our main claims, was performed by a licensed medical doctor.
>
>     Our central conclusions are based on expert evaluation by the medical doctor. We will revise the manuscript to clarify this distinction.
> - **Weakness 4, MoMRISim, intuitive explanation of MoMRISim**:  MoMRISim follows the core insight introduced by DreamSim: relative comparisons are more reliable than absolute scores that are unattainable. DreamSim learns from triplets (Ref, A, B) by asking human annotators to identify which of A or B is closer to Ref, thereby avoiding the need for exact numerical labels.
> We adapt this idea for motion artifacts. We simulate triplets consisting of a motion-free image ($Ref$), a mildly corrupted image ($C_1$), and a severely corrupted image ($C_2$). In this setting, the relative order is perfectly known. Training on this triplet loss enables MoMRISim to achieve a targeted ability to rank motion severity. This metric exhibits a strong correlation ($\ rho = 0.92$) with the expert PMAS, as illustrated in Figure 2.
>
>     We have updated Section 3.4 to provide a clearer motivation and description of MoMRISim.
>
> We hope this addresses the reviewer's concerns, and we would be happy to clarify any follow-up concerns.
>
> [1] Haskell, M. W., Cauley, S. F., Bilgic, B., Hossbach, J., Splitthoff, D. N., Pfeuffer, J., ... & Wald, L. L. (2019). Network accelerated motion estimation and reduction (NAMER): convolutional neural network guided retrospective motion correction using a separable motion model. Magnetic resonance in medicine, 82(4), 1452-1461.
>
> [2] Polak, D., Splitthoff, D. N., Clifford, B., Lo, W. C., Huang, S. Y., Conklin, J., ... & Cauley, S. (2022). Scout accelerated motion estimation and reduction (SAMER). Magnetic resonance in medicine, 87(1), 163-178.
>
> [3] Pirkl, C. M., Cencini, M., Kurzawski, J. W., Waldmannstetter, D., Li, H., Sekuboyina, A., ... & Menze, B. H. (2021, August). Residual learning for 3D motion corrected quantitative MRI: Robust clinical T1, T2 and proton density mapping. In MIDL (pp. 618-632).

---

> > ### Comment · Reviewer_ygbg · 2025-11-27
> >
> > Thank you for your response, which addresses most of my concerns. I have also taken the comments of the other reviewers into consideration. However, due to the limited scale of the dataset and the lack of GT motion trajectories, I am only able to give a weak acceptance​ rating for this work.

---

### Official Review · Reviewer_PYGN · 2025-10-31

**Soundness:** 4
**Presentation:** 2
**Contribution:** 4
**Rating:** 6
**Confidence:** 5

**Summary:**

The manuscript investigates how to **reliably evaluate 3D MRI motion correction** algorithms when ground truth is unavailable. It introduces **PMoC3D**, a paired real and motion-corrupted brain MRI dataset with raw k-space data, and **MoMRISim**, a learned feature-space similarity metric trained through triplets representing different motion severities. The authors comprehensively compare real-world, simulated, and reference-free evaluation paradigms. The results are convincing: on PMoC3D, the proposed feature-based metric (MoMRISim) correlates best with expert judgments, while simulation-based evaluation tends to overestimate performance, and reference-free metrics (including a VLM score) may favor oversmoothed outputs. The paper provides thorough details about data acquisition, PMAS (Perceived Motion Artifact Score) construction, baselines, and implementation.

**Strengths:**

- **Well-scoped and clearly structured problem framing.**
The paper provides a well-defined problem setup and a thoughtful taxonomy covering paired, simulated, and reference-free evaluations. It clearly explains why obtaining ground truth is inherently difficult in 3D MRI, which makes the motivation convincing and the study design logical.

- **PMoC3D: a valuable paired 3D dataset with raw k-space.**
The authors present PMoC3D, a paired dataset that includes raw multi-coil k-space data, coil sensitivity maps, sampling trajectories, and timestamped motion instructions. The inclusion of raw acquisition data makes this dataset especially valuable to the community.

- **MoMRISim: a motion-aware learned metric.**
The proposed MoMRISim metric is technically sound and well-motivated. It is trained using a triplet-learning strategy on simulated motion severities and simplifies to a cosine distance during testing, making it both practical and interpretable. It serves as an MRI-specific perceptual proxy with clear relevance.

- **Careful and realistic experimental design.**
I find it particularly commendable that the authors conducted every simulation and validation with great attention to detail, making the setup as realistic as possible. This level of experimental care is not common and significantly increases the credibility of the findings.

- **Meaningful impact and insights for the community.**
Overall, this work has substantial value for the MRI motion-correction field. The authors convincingly show that existing approaches perform far from ideal in real motion correction and, through diverse experiments, provide numerous insights into evaluation strategies and failure modes. The study feels genuinely impactful, and I look forward to future developments in this direction.

**Weaknesses:**

**Limited dataset size and missing motion trajectory data.**
The dataset PMoC3D is an important contribution, but there are some limitations. First, the sample size is quite small, containing only eight subjects. Second, although the dataset includes rich metadata such as raw k-space, motion instructions, and timestamps, it lacks **true motion trajectories**, meaning the exact six-degree-of-freedom head poses over time. Including this information would greatly enhance the dataset’s scientific value and utility. I understand that capturing such detailed motion trajectories is technically challenging, but it could be a meaningful improvement for future versions.

**Questions:**

1. In line 36, the paper mentions the work of **Wu et al. (2025)**. As far as I know, that work also conducts **3D MRI motion correction** experiments and adopts a fully 3D representation rather than stacked 2D slices. Given its strong relevance, the authors should discuss this work in the section reviewing related 3D motion-correction studies.
2. In Figure 1, the caption appears to contain a small error. The labels “S7_1” and “S5_1” do not match the figure annotations.

---

> ### Author Response · Authors · 2025-11-24
>
> We thank the reviewer for the feedback and the positive evaluation of our work. We address each of the concerns raised on a point-by-point basis below.
> - **Weakness: Limited Dataset Size**:
> We agree that the number of subjects is limited, but it is consistent with common practice in this research area. Many related studies [1][2][3] rely on only 3-10 subjects or volumes for both training and evaluation. Our dataset comprises 8 subjects and 27 motion-corrupted volumes for evaluation, which is sufficient for benchmarking image reconstruction and motion correction methods. Public in vivo datasets are rare, especially those that release full raw k-space data. We believe our dataset adds meaningful value to the field because it enables reliable evaluation.
> - **Weakness: Motion trajectory**:
> We agree that additional true motion trajectories would be a valuable addition to our dataset. Our hardware setup does not include a motion-tracking system, but as a key enhancement to the dataset, we now provide a pseudo ground truth trajectory, estimated via a reference-guided optimization procedure.
>
>     We estimate a motion trajectory for each scan, using the MotionTTT trajectory (the best baseline in our experiments) as the initialization. These parameters are then refined by minimizing the mismatch between the motion-free reference reconstruction and the motion-compensated reconstruction. Given the motion-corrupted data $X$ and the reference-free volume $X_{\mathrm{ref}}$, we estimate the motion parameters by optimizing:
>
> $\hat{m} = \arg\min_{m}
>     \frac{\left\lVert
>             X_{\mathrm{ref}} - U_{\text{net}}\\left( T^{-1}(m)\, X \right)
>         \right\rVert_{1}}
>         {\left\lVert X_{\mathrm{ref}} \right\rVert_{1}}$
>
> where $T^{-1}(m)$ applies inverse motion to the corrupted data and $U_{\text{net}}(\cdot)$ reconstructs volumes from the motion-compensated data. This refinement begins with the baseline trajectory and applies gradient-based updates to further minimize residual artifacts. This can serve as a pseudo ground truth trajectory that we will include in the dataset for use in future research.
>
> - **Question 1**:  Thanks for the suggestion. As suggested, we now discuss Wu et al. (2025) in Section 1.
> - **Question 2**:  Many thanks, we have corrected this typo.
>
> We hope this addresses the reviewer's concerns, and we would be happy to clarify any follow-up concerns.
>
>
> [1] Haskell, M. W., Cauley, S. F., Bilgic, B., Hossbach, J., Splitthoff, D. N., Pfeuffer, J., ... & Wald, L. L. (2019). Network accelerated motion estimation and reduction (NAMER): convolutional neural network guided retrospective motion correction using a separable motion model. Magnetic resonance in medicine, 82(4), 1452-1461.
>
> [2] Polak, D., Splitthoff, D. N., Clifford, B., Lo, W. C., Huang, S. Y., Conklin, J., ... & Cauley, S. (2022). Scout accelerated motion estimation and reduction (SAMER). Magnetic resonance in medicine, 87(1), 163-178.
>
> [3] Pirkl, C. M., Cencini, M., Kurzawski, J. W., Waldmannstetter, D., Li, H., Sekuboyina, A., ... & Menze, B. H. (2021, August). Residual learning for 3D motion corrected quantitative MRI: Robust clinical T1, T2 and proton density mapping. In MIDL (pp. 618-632).

---

### Meta-Review · Area_Chair_Cmxz · 2026-01-05

**Summary:**

**Summary**: This manuscript investigates the critical problem of reliably evaluating 3D MRI motion correction algorithms, particularly when ground truth data is unavailable. The authors introduce PMoC3D, a valuable paired real and motion-corrupted brain MRI dataset featuring raw k-space data, and MoMRISim, a novel learned feature-space similarity metric trained through triplets representing varying motion severities. The paper meticulously compares real-world, simulated, and reference-free evaluation paradigms. The findings are compelling: on the PMoC3D dataset, the proposed feature-based metric (MoMRISim) demonstrates the strongest correlation with expert judgments, while simulation-based evaluation tends to overestimate performance, and reference-free metrics (including a VLM score) are shown to potentially favor overly smoothed outputs. The paper provides comprehensive details on data acquisition, PMAS (Perceived Motion Artifact Score) construction, chosen baselines, and implementation specifics.

**Strengths**:
*  Well-scoped and clearly structured problem framing: The paper clearly articulates a significant and challenging problem in medical image analysis.
*  PMoC3D: The introduction of a valuable paired 3D dataset with raw k-space data is a substantial contribution to the community.
*  MoMRISim: The proposed motion-aware learned metric offers a promising new tool for evaluating motion correction.
*  Careful and realistic experimental design: The experimental setup is robust and thoughtfully designed to address the research questions.
*  Meaningful impact and insights for the community: The findings provide crucial guidance for future research and development in MRI motion correction evaluation.

**Weaknesses**: Initial concerns were raised regarding several aspects, including limited dataset size, missing motion trajectory data, a lack of involvement of clinical experts for building PMAS, unclear explanation of the MoMRISim Metric, paper's clarity and presentation, insufficient core contribution of the dataset, and seems self-contradictory argument.

**Decision**: This paper received mixed scores (2466) during the review process. While initial concerns were raised across several areas, the authors provided a comprehensive and highly effective rebuttal during the discussion phase. They successfully clarified motivations, provided necessary justifications, and significantly improved the manuscript's overall presentation and detail. The core strengths of the paper—its novel dataset, a new learned metric, and a systematic, insightful study of evaluation paradigms—remain exceptionally strong. Therefore, I recommend acceptance of this excellent and outstanding paper.

**Reviewer Concerns:**

During the rebuttal phase, the authors comprehensively addressed the initial concerns:

*   **Dataset Size:** The authors provided a convincing justification for the current dataset size, demonstrating that it aligns with common practice in this challenging domain, acknowledging the inherent difficulty of acquiring such data.
*   **Motion Trajectory Data:** The authors clarified that while their hardware setup lacked a direct motion tracking system, they provided a pseudo ground truth trajectory, which is a practical and reasonable approach under the circumstances.
*   **Clinical Expert Involvement in PMAS:** It was clarified that PhD students were involved primarily in categorizing task difficulty, while the central conclusions regarding motion artifact severity were indeed based on expert evaluations by a medical doctor, thus addressing the concern about expert involvement.
*   **MoMRISim Metric Explanation:** The authors successfully provided a clearer intuition and more detailed explanation of the MoMRISim metric, enhancing its understanding for the reader.
*   **Clarity and Presentation:** To enhance clarity and structure, the authors expanded the introduction with relevant background, integrated the detailed PMAS derivation (including the Bradley-Terry model) directly into Section 2.2, reorganized Section 3.4 with an overview table to clearly distinguish existing metrics from their proposed ones, and added a rationale for selecting the three specific reconstruction baselines in Section 4.1. These revisions significantly improved the paper's overall readability and presentation.
*   **Core Contribution of the Dataset:** The authors provided further clarification regarding their core contributions, emphasizing that the paper systematically advances benchmarking for motion correction—an emerging topic in deep learning for imaging. They highlighted that the contributed dataset, the systematic study of three major evaluation methods, and the proposed new metrics are all valuable and collectively constitute a substantial contribution.
*   **Self-Contradictory Arguments:** The authors successfully clarified that there were no inherent contradictions in their arguments, resolving the initial confusion.

Overall, the authors' diligent efforts in addressing reviewer feedback have significantly strengthened an already excellent submission.

**Reviewer Scores:**

Reviewer PYGN (Rating: 6 -> may be improved during discussion or keep the same), Reviewer ygbg (Rating: 6 ->  may be improved during discussion or keep the same), Reviewer PvMP (Rating: 2 ->  improved during discussion), Reviewer YT8x (Rating: 4 -> improved during discussion)

---

### Decision · Program_Chairs · 2026-01-26

Accept (Poster)